# RiverBedDynamics v1.0: A Landlab component for computing two-dimensional sediment transport and river bed evolution

Angel D. Monsalve[1], Samuel R. Anderson[2], Nicole M. Gasparini[2], Elowyn M. Yager[1]

[1]Center for Ecohydraulics Research, Department of Civil and Environmental Engineering, University of Idaho, Boise, ID, USA
[2]Department of Earth and Environmental Sciences, Tulane University, New Orleans, LA, USA

*Correspondence to*: Angel Monsalve (amonsalve@uidaho.edu)

**Abstract.** Computational landscape evolution models (LEMs) typically comprise at least two interacting components: a flow hydraulics solver that routes water across a landscape and a fluvial geomorphological model that modifies terrain properties, primarily bed surface elevation. LEMs used in long-term simulations over large watersheds, including some available in the Landlab library, often assume that only erosive processes occur in rivers and that terrain elevation increases solely due to tectonic uplift. Consequently, these models cannot capture the dynamics of gravel-bedded rivers, lacking the capacity to include sediment mixtures, simulate sediment deposition, and track textural changes in substrate stratigraphy that result from varying flow characteristics. To address this limitation, we developed, implemented, and tested RiverBedDynamics, a new Landlab component that simulates the evolution of bed surface elevation and grain size distribution in two-dimensional grids based on the Exner equation for sediment mass balance. By dynamically coupling RiverBedDynamics with Landlab's hydrodynamic flow solver, OverlandFlow, we created a new LEM capable of simulating the dynamics of local shear stresses, bed load transport rates, and grain size distributions. Comparisons of our LEM results with analytical and previously reported solutions demonstrate its ability to accurately predict time-varying local changes in bed surface elevation, including erosion and deposition, as well as grain size distribution. Furthermore, application of our LEM to a synthetic watershed illustrates how spatially variable rainfall intensity leads to varying discharge patterns, which in turn drive changes in bed elevation and grain size distribution across the domain. This approach provides a more comprehensive representation of the complex interactions between flow dynamics and sediment transport in gravel-bedded rivers at timescales ranging from individual flood events to yearly morphological changes, enhancing our ability to model landscape evolution across diverse geomorphic settings.

# 1 Introduction

Landscape Evolution Models (LEMs) are fundamental tools for geomorphologists allowing researchers to understand landscape morphology produced under different climatic and tectonic circumstances, and in some cases, inform management decisions (Coulthard, 2001). These models simulate the long-term development of landscapes by incorporating multiple
geomorphic processes, providing insights into how terrains change over time under various environmental conditions. To achieve this, currently available LEMs differ in the number of physical processes considered, the way they route water and sediment across a landscape, and how the domain and its features are represented when solving the governing equations (Coulthard, 2001; Temme et al., 2017; Tucker and Hancock, 2010).

While many LEMs have contributed significantly to our understanding of landscape evolution, they often rely on simplifications that limit their applicability to certain geomorphic contexts. When designed to simulate long time periods, these models incorporate significant assumptions to make computations feasible. These simplifications often include steady-state flow assumptions, omission of grain size variations, and in many cases, exclusion of sediment deposition processes. A common assumption is that erosive river processes and tectonic uplift are the primary factors shaping a landscape over long
time scales (Campforts et al., 2017; Forte et al., 2016; Langston and Tucker, 2018; Li et al., 2018; Whipple et al., 2017). For instance, one of the earlier LEMS, GOLEM, (Tucker and Slingerland, 1994) assumed a steady single flow direction with water discharge defined as the product of drainage area and rainfall rate. This approach is still common in LEMs because it greatly simplifies calculations (e.g., Braun & Willett, 2013; Campforts et al., 2017; Goren et al., 2014; Mitchell & Forte, 2023; Tucker et al., 2001). In contrast, CAESAR (Coulthard et al., 2002) employs a routing scanning algorithm that allows
multiple flow directions and incorporates flow variability, which enables the simulation of non-steady state flow conditions and flood wave routing.

To address these limitations, more recent frameworks have focused on advancing sediment transport and deposition modeling capabilities. For instance, CIDRE (Carretier et al., 2023) employs a Lagrangian framework with size-dependent
sediment deposition probability, while River.lab (Davy et al., 2017) implements an Eulerian approach. (Le Minor et al., 2022) further enhanced these capabilities through improved deposition modeling approaches. While these frameworks have successfully captured key aspects of sediment transport and deposition, they lack the flexibility and comprehensive grain size tracking capabilities that we introduce with RiverBedDynamics. Our component offers unique advantages through its seamless integration with the Landlab platform, allowing easy coupling with any flow solver, and its comprehensive tracking
of temporal grain size distribution evolution.

The evolution of landscape modeling frameworks, from earlier LEMs to recent sediment transport solutions, highlights fundamental challenges in simulating river systems across different spatial and temporal scales. In terms of representing

drainage networks and channels within a catchment, LEMs employ various strategies to address the challenge of scale. This is crucial because the model grid resolution significantly impacts how different landscape features—such as channels, floodplains, and hillslopes—are represented and captured by the model, and consequently, how governing equations are solved. For example, the CHILD model (Tucker et al., 2001) uses an adaptive triangulated irregular mesh to accurately capture transitions between different landscape elements, particularly the boundaries between channels and floodplains. This approach allows for a more detailed representation of channel networks, a characteristic that is not achievable when using uniform rectangular grid elements. In contrast, CAESAR employs a finer grid resolution near and within channels, especially at their boundaries to capture the different elements. This method effectively concentrates computational resources where they are most needed for precise flow and sediment transport calculations. The choice of model complexity and resolution has significant implications for the timescales over which LEMs can operate effectively. Simpler models with more assumptions can simulate geomorphological changes over millennia to millions of years with relatively short computation times, but at the cost of reduced accuracy and precision. On the other hand, more detailed models like SedFoam (Cheng et al., 2017) for instance, can predict small-scale phenomena such as sand concentrations in the water column at a submillimeter scale, but are computationally expensive and typically limited to shorter timescales and smaller spatial extents.

Beyond considerations of scale and resolution, a critical challenge in landscape evolution modeling is accurately representing the physical processes in gravel-bedded rivers. Current LEMs have been particularly successful in modeling bedrock channels, where the rate of sediment removal is limited by the detachment of material from the bed (detachment-limited conditions). However, this simplification is not adequate when applied to gravel-bedded rivers, where the rate of sediment transport is limited by the flow's capacity to move sediment (transport-limited conditions) (e.g., Attal et al., 2011; Gasparini et al., 2004; Whipple & Tucker, 2002). The evolution of alluvial channel geometries in gravel-bed rivers depends on both erosion and deposition, processes intricately linked to grain size distributions and their evolution over time. Very few LEMs include explicit treatment of grain sizes, yet this factor is critical for both accurately simulating sediment transport, channel morphodynamics, and understanding the formation of fluvial deposits. The ability to track grain size evolution is essential for modeling realistic sediment transport rates, while also enabling investigation of stratigraphic development and preservation in river systems. Moreover, linking grain size distributions with hydrograph variations is key to modeling realistic shear stress values and provides insights into both the immediate channel response and the longer-term depositional record.

Incorporating deposition alongside erosion in LEMs as a mass conservation problem introduces additional complexity to model development. This approach requires conducting a mass balance at individual cells or control volumes, significantly increasing the computational demands of the model. Thus, LEMs that have adapted a mass balance approach have generally relied on simplified hydrology (e.g., Attal et al., 2011; Gasparini et al., 2004; Whipple & Tucker, 2002). These studies prioritize large-scale trends over long time scales, rather than exploring detailed bed evolution. However, accurately

modeling erosion and deposition of different grain sizes remains crucial for understanding gravel-bed river dynamics. This requires a more accurate representation of flow velocity and depth, as these variables are used for calculating sediment

transport capacity and determining whether deposition or erosion occurs at any given location.

The need for new models to accurately model grain size distributions presents a significant opportunity for advancing our understanding of landscape evolution, particularly in the context of gravel-bed rivers. Gravel-bed rivers are of paramount importance in geomorphology due to their role in sediment routing, their sensitivity to changes in sediment supply and flow

regimes, and their close coupling with hillslope processes. Developing a model that can simulate the evolution of gravel-bed streams in a continuum framework, while also linking this evolution to broader landscape processes, would represent a major step forward in the field.

To address these modeling challenges, we have developed RiverBedDynamics, a component with a flexible design that

operates across various spatial and temporal scales. The component is particularly well-suited for analyzing relatively short-term processes (hours to years), where detailed grain-scale interactions and non-steady flow conditions are important. Although theoretically capable of handling longer timescales, the component's mechanistic approach to sediment transport and explicit treatment of flow hydraulics makes it computationally intensive for simulations spanning decades to centuries. The spatial resolution can range from fine-scale process representation (meters) to watershed-scale applications (tens of

meters), with users needing to balance tradeoffs between spatial resolution, computational demands, and physical process representation.

This new modeling capability enables researchers to address several important questions in fluvial geomorphology, such as: (i) How do variations in sediment supply and grain size distributions influence channel morphodynamics during flood

events? (ii) What are the feedbacks between bed surface texture evolution and channel adjustment during unsteady flows? (iii) How do spatial patterns of erosion and deposition respond to changing climate and land use conditions across watershed scales? (iv) What is the role of grain sorting processes in determining channel stability and sediment connectivity through river networks? and (v) How are climate-driven hydrologic variations preserved in fluvial stratigraphic records, and what can these records tell us about past environmental conditions?


To develop a modeling framework capable of addressing these questions, we leveraged recent advancements in computational platforms. One such framework is Landlab, a Python-based platform designed for creating, assembling, and running 2D landscape evolution models (Barnhart et al., 2020; Hobley et al., 2017). Landlab's modular structure allows researchers to combine various components, each representing different geomorphic processes, to create customized LEMs

tailored to specific research questions. Recent efforts within the Landlab framework have made significant strides in improving the accuracy of flow routing and erosion modeling. For instance, Adams et al. (2017) coupled the OverlandFlow

component with DetachmentLtdErosion to investigate watershed incision patterns. The OverlandFlow component simulates surface water flow across a landscape, while DetachmentLtdErosion models the erosion of bedrock or cohesive sediment. However, while this approach represents an improvement in flow routing accuracy, it cannot be directly used in modeling

gravel-bedded rivers. The DetachmentLtdErosion component, based on the stream power law, does not account for the complex dynamics of sediment transport and deposition characteristic of gravel-bed systems. Specifically, it doesn't consider the movement of different grain sizes or the process of sediment deposition, both of which are crucial in gravel-bed river evolution.

An alternative approach within Landlab is the NetworkSedimentTransporter component (Pfeiffer et al., 2020). This Lagrangian model predicts changes in bed material grain size and river bed elevation based on bed load estimates within a predefined river network. It provides valuable insights for long-term simulations of sediment dynamics in established channel networks. While optimized for efficiency in predefined river networks, the NetworkSedimentTransporter was designed with different goals than our component. It operates with a static channel configuration throughout the simulation

and focuses on efficient computation rather than tracking stratigraphic evolution or handling unsteady flow conditions. The component was purposefully engineered to process sediment through discrete packets, which offers computational advantages for certain applications. Our RiverBedDynamics component complements this existing work by addressing different research questions, particularly those requiring continuous grain size distributions and dynamic responses to varying hydrological conditions.


The development of these components within Landlab has progressively enhanced our ability to model different aspects of landscape evolution. However, there remains a need for a component that can simulate the full range of processes occurring in gravel-bed rivers while taking advantage of Landlab's integrative framework. While our component primarily addresses sediment transport by flowing water—applying these physics across both channelized and unchannelized areas during

overland flow events—other geomorphic processes such as soil creep, landsliding, and bank erosion require different physical representations through specialized Landlab components. In particular, the ability to model sediment transport and deposition across varying flow conditions, while tracking both spatial and temporal evolution of grain size distributions, would represent a significant advance in our modeling capabilities.

These existing components, while valuable, highlight a critical gap in our ability to model gravel-bed rivers within the context of landscape evolution. To address these limitations, a component is required that can: i) Accurately represent sediment transport dynamics in gravel-bedded rivers, ii) Account for fractional sediment transport including erosion and deposition processes, iii) Predict bed surface changes across an entire watershed, and iv) Integrate with high-accuracy flow prediction under non-steady and non-uniform conditions.


To address this gap, we developed RiverBedDynamics, a new Landlab component designed to simulate the evolution of gravel-bed streams in a continuum model, allowing for the integration of the channel with other geomorphological processes using the Landlab platform. By coupling RiverBedDynamics with OverlandFlow, we enable the simulation of non-steady flow conditions, which is crucial for understanding the complex dynamics of gravel-bed rivers. The component addresses the limitations of existing models by incorporating grain size evolution, non-steady flow conditions, and watershed-wide predictions of bed surface changes. A key feature of our component is its ability to track and preserve stratigraphic information, enabling investigations of both short-term morphodynamics and longer-term geological processes. This stratigraphy tracking functionality allows researchers to study how temporal variations in flow and sediment transport conditions are recorded in depositional sequences, providing insights into the geological history of river systems. This advancement enables researchers to address several important questions in fluvial geomorphology, such as: (i) How do variations in sediment supply and grain size distributions influence channel morphodynamics during flood events? (ii) What are the feedbacks between bed surface evolution and flood wave propagation through river networks? (iii) How do spatial patterns of erosion and deposition respond to changing climate and land use conditions across watershed scales? and (iv) What is the role of grain sorting processes in determining channel stability and sediment connectivity through river networks?

In this article, we introduce RiverBedDynamics and demonstrate its capabilities in modeling gravel-bed river evolution within a landscape context. Through a series of tests and applications, we illustrate how RiverBedDynamics can capture key aspects (e.g., sediment transport dynamics, grain size evolution, erosion and deposition processes, and watershed-scale bed surface changes) of landscape evolution, particularly in systems dominated by gravel-bed rivers. All sediment transport predictions are based on the unsteady total shear stress, which accounts for spatial and temporal gradients in flow velocity and local variations in bed elevation and water depth. Evaluations of our component are conducted using test cases with analytical solutions from previously available models. An example in a large watershed is used to explore large scale applications of the component.

**2 A general overview of the Landlab modeling approach**

Landlab, a Python-based interdisciplinary open-source platform, serves as a robust framework for developing computational landscape models addressing earth surface dynamic processes (Barnhart et al., 2020; Hobley et al., 2017; Tucker et al., 2022). We integrate our component into Landlab, leveraging its seamless support for incorporating new process components. Further, Landlab already contains a simplified hydrodynamic model for computing flow variables, upon which RiverBedDynamics relies (Adams et al., 2017). Numerous studies detail the general structure of Landlab (e.g., Adams et al., 2017; Barnhart et al., 2019, 2020; Hobley et al., 2017; Shobe et al., 2017; Tucker et al., 2022), therefore, we focus here solely on aspects relevant to implementing the RiverBedDynamics component.

At the core of our component lies Landlab's gridding engine, facilitating data manipulation and exchange among various

components. This engine operates on a 2D structured grid, enabling numerical operations essential for calculating flow, bed surface changes, and sediment variables during simulations. For instance, the grid contains methods for computing topographic gradients, sediment mass balance, and mapping velocities from grid links to nodes. Currently, our component exclusively operates on raster grids (Figure 1). Within this grid framework, nodes represent discrete (x;y) points, while links denote vectors connecting neighboring nodes with fixed directionality. A cell, bounded by faces, encapsulates the area

around a non-perimeter (i.e., interior) node. All cells within our component are rectangular-shaped and maintain uniform dimensions in both the x ($\Delta x$) and y ($\Delta y$) directions.

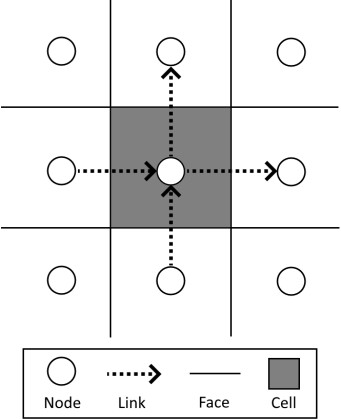

Figure 1: Elements of a Landlab grid used by our component, illustrating a typical grid segment. Information is stored in
nodes and links. For instance, surface bed elevation data is held at the nodes, whereas the gradients of these elevations are stored in the links.

Our component was developed around the RasterModelGrid class, chosen for its ability to facilitate the numerical solution of
partial differential equations, such as the Exner equation for sediment mass conservation, and spatially variable processes in a straightforward manner. For instance, consider the Meyer-Peter & Müller (1948) equation for bed load transport:

$$q_b^* = \begin{cases} 8(\tau^* - \tau_c^*)^{3/2} , \tau^* > \tau_c^* \\ 0 , \tau^* \leq \tau_c^* \end{cases} \qquad \text{Eq. 1}$$

where $\tau_c^* = 0.047$ is the dimensionless critical shear stress, $q_b^*$ is the dimensionless volumetric bed load transport rate per unit width, and $\tau^*$ is the dimensionless shear stress. Implementation of Eq. 1 requires calculating $\tau^*$ at each node and determining locations where $\tau^*$ exceeds $\tau_c^*$ to compute $q_b^*$. Using Landlab's structure and tools, Eq. 1 can be implemented as:

*mask = tau_star > tau_star_cr*

*qb_star[mask] = qb_star_coeff * (tau_star[mask] - tau_star_cr[mask]) ** (qb_star_exp)*

here, *tau_star* is extracted from the grid, while *tau_star_cr , qb_star_coeff,* and *qb_star_exp* are the equation's parameters (

0.047, 8, and 3/2, respectively). This implementation illustrates a key advantage of Landlab's design: the ability to translate

mathematical equations directly into code that operates efficiently across the entire domain. The similarity between Eq. 1

and its implementation demonstrates how Landlab's grid-based structure eliminates the need for explicit iteration over

individual cells, making the code both intuitive and computationally efficient. This direct mapping between mathematics and

code is a fundamental feature that extends to all processes implemented in the framework.. Additionally, data exchange

between different Landlab components is seamlessly facilitated by the grid. For instance, modifications to the bed surface

elevation in RiverBedDynamics result in the updated value being immediately available to all Landlab components via the

field *grid["node"]["topographic__elevation"]*.

Boundary conditions in our component are inherited from the Landlab grid object, aligning with those specified in the

OverlandFlow component (Adams et al., 2017). Nodes defined as 'boundary' can be designated as either open, fixed

gradient, or closed. Open boundary nodes allow flux to enter or leave the model domain, acting as flow outlets. Closed

boundary nodes prevent flux from entering or leaving the domain. This classification determines the behavior of surface

water flow at these boundary nodes, with sediment fluxes calculated based on local flow conditions. Links connected to

these boundary nodes are automatically classified as active, inactive, or fixed. Active links, where fluxes are calculated,

occur between two core nodes or between a core node and an open boundary node. Inactive links, where no fluxes are

calculated, occur between a closed boundary node and a core node or between any pair of boundary nodes. Fixed links,

which can have assigned values, occur between a fixed gradient node and a core node. The sole exception to these inherited

conditions is at the domain outlet. Here, RiverBedDynamics requires specifying either a fixed bed surface elevation or a

zero-gradient condition (refer to section 3.4 for more details).

## 3 Model description

Our component was designed to work in conjunction with a flow solver such that continuous feedback between surface flow

and river bed properties determines the behavior of the system. While RiverBedDynamics is compatible with any flow solver

through Landlab's "plug-and-play" capabilities, in the examples presented in this paper, we demonstrate its use with the

OverlandFlow component (Figure 2). This specific choice of flow solver illustrates one possible implementation, but users

can integrate RiverBedDynamics with other flow routing components available in Landlab depending on their needs. At

each time-step, the flow governing equations are solved across the entire domain by OverlandFlow, obtaining flow depth,

velocity, and discharge. The routines of RiverBedDynamics can be conceptualized in two major parts: i) bed load transport

and ii) river bed evolution. In the first part, RiverBedDynamics processes surface flow and bed surface grain size properties

stored in the grid to calculate local shear stress and bed load transport rate. In the second part, it uses sediment fluxes

entering and leaving each cell to compute the mass balance. This process updates the bed surface elevation and bed properties, such as grain size distribution, thereby completing the cycle at each time step.

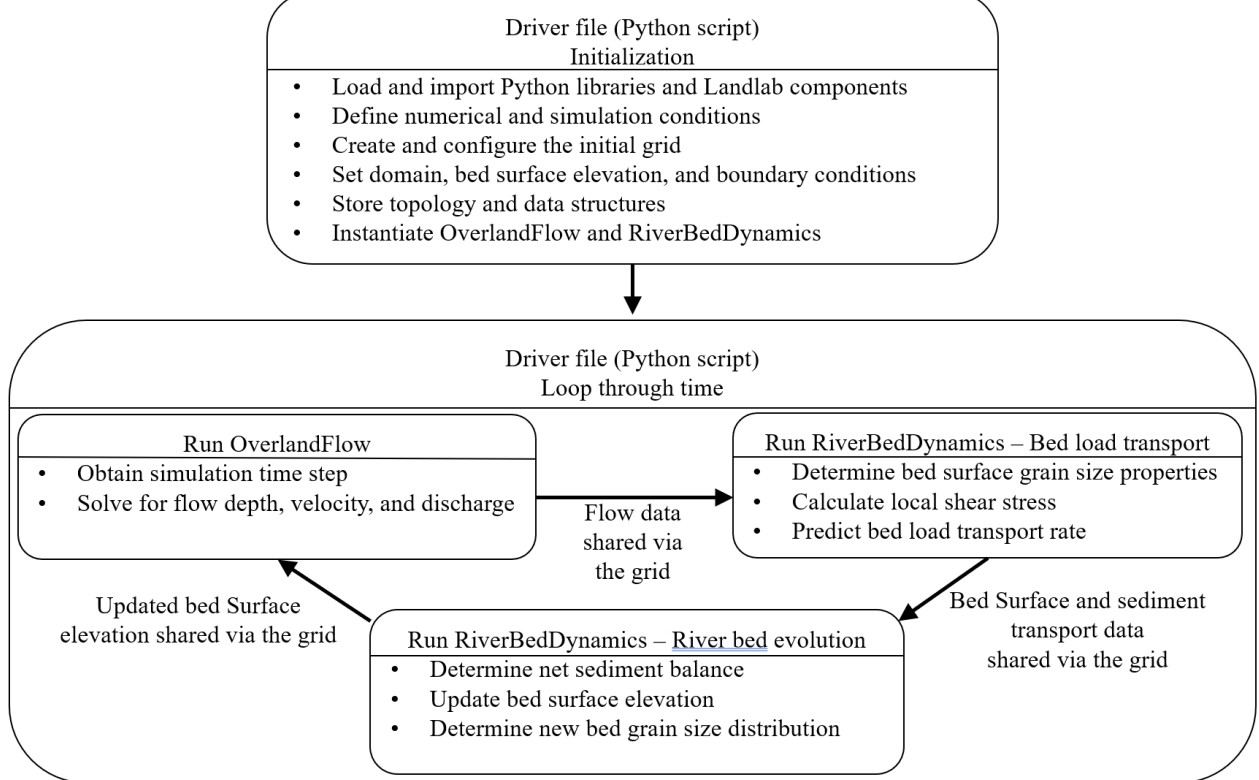

Figure 2: Simplified workflow for the coupled OverlandFlow and RiverBedDynamics routine. The driver file is a procedure script containing the set of instructions to create all the required data and loop through time, dynamically linking and updating surface flow and river sediment variables.

## 3.1 Flow variables and shear stress calculations

During each time step of a simulation, OverlandFlow solves the 2D flow equations across all grid links determining the surface water discharge per unit width ($q$), flow velocity components ($u$ and $v$, corresponding to the x and y directions , respectively) and water depth ($h$) at links. Subsequently, water depth at nodes is calculated based on mass conservation, factoring in all inflow and outflow at a given node. These spatial distributions of flow variables provide the foundation for subsequent sediment transport calculations. While flow velocity is not directly derived, it can be calculated at links according to $U = q/h$ with velocity components $u$ and $v$ for x and y directions, respectively. Our sediment transport rate calculations are based on the local shear stress considering an unsteady friction slope (Ghimire and Deng, 2011) according to:

$$S_{fx} = -\left(\frac{\partial \eta}{\partial x} + \frac{\partial h}{\partial x} + \frac{u}{g}\frac{\partial u}{\partial x} + \frac{u}{g}\frac{\partial u}{\partial t}\right) \quad \text{and} \quad S_{fy} = -\left(\frac{\partial \eta}{\partial y} + \frac{\partial h}{\partial y} + \frac{v}{g}\frac{\partial v}{\partial y} + \frac{v}{g}\frac{\partial v}{\partial t}\right)$$

Eq. 2

where $u$ and $v$ are the velocity components in the x and y directions, respectively; $S_{fx}$ and $S_{fy}$ are the friction slopes evaluated in the x and y directions, respectively; η is the bed surface elevation; $g$ is the acceleration due to gravity; and $t$ is time. This formulation is an extension used in shear stress calculations of the commonly used depth-slope product, in which steady and uniform flow conditions are assumed, resulting in $S_{fx} = -\partial \eta/\partial x$ in the $x$ direction. The unsteady friction slope formulation accounts for both spatial variations in flow properties and their temporal changes, including acceleration and deceleration effects that are particularly important during flood events.

Each individual term in Eq. 2 is calculated directly using built-in methods of the Landlab grids. Topographic gradients ($\partial \eta/\partial x$ and $\partial \eta/\partial y$) are based on the bed elevation slope at the nodes defining a link using the *calc_grad_at_link* method. The same approach is used for water depth spatial gradients ($\partial h/\partial x$ and $\partial h/\partial y$). Velocity spatial gradients are approximated using a central difference scheme according to:

$$\frac{\partial u}{\partial x} \approx \frac{u_{x+1} - u_{x-1}}{2\Delta x} \quad \text{and} \quad \frac{\partial v}{\partial y} \approx \frac{v_{y+1} - v_{y-1}}{2\Delta y}$$

Eq. 3

where subscripts $x+1$, $x-1$, $y+1$, and $y-1$ indicate to the right, left, above, and below, respectively, the location of the link considered (Figure 3). Velocity time gradients are approximated using the backward Euler method defined as:

$$\frac{\partial u}{\partial t} \approx \frac{u^t - u^{t-1}}{\Delta t} \quad \text{and} \quad \frac{\partial v}{\partial t} \approx \frac{v^t - v^{t-1}}{\Delta t}$$

Eq. 4

where $\Delta t$ is the time step and the superscripts $t$ and $t-1$ indicate the current and previous time steps. The velocity at the previous time step $(t-1)$ is stored in the *field surface_water__velocity_prev_time_link* and is updated every time step. At the beginning of the simulation, this gradient can be forced to zero or initialized with a known velocity value if available as an initial condition.

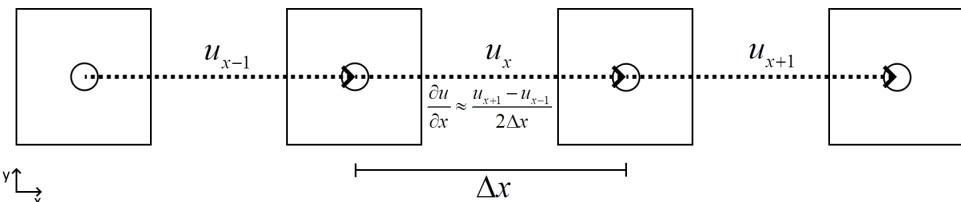

Figure 3: A representation of the stencil used to calculate the velocity gradient at links. Cells are separated only to highlight the definition of velocities at links. The gradient $\partial u/\partial x$ at the location of the link with velocity $u_x$ is estimated using a central difference scheme and considers the neighboring links. The same principle applies to calculate $\partial v/\partial y$ but is not represented in this figure.

While a two-point scheme, like upwind, provides better numerical stability in advection-dominated flows, we opted for the central difference scheme primarily because it provides more accurate approximations of spatial gradients (with errors proportional to the square of the grid spacing) compared to first-order schemes like upwind differencing, and because of its ability to capture multi-directional information propagation in our coupled system. This choice reflects a balance between computational accuracy and physical representation, particularly important when simulating complex watershed-scale processes with varying flow conditions. The scheme's higher-order accuracy and multi-directional nature are especially beneficial for resolving flow dynamics at channel confluences and during rapid flow variations, where information propagates in multiple directions simultaneously.

The local shear stress at each link is then calculated according to:

$$\tau_x = \rho g h S_{fx} \text{ or } \tau_y = \rho g h S_{fy} \qquad\qquad \text{Eq. 5}$$

Typically, shear stress in river channels is defined using the hydraulic radius rather than water depth to account for bank roughness. However, the choice between these approaches depends on the spatial resolution relative to channel width and the processes being represented. We use water depth as the default option for calculating shear stress because: i) When cell sizes are much smaller than channel widths, most cells represent the channel interior and do not contain river banks, making the hydraulic radius effectively equivalent to water depth, ii) When cell sizes are much larger than channel widths, a single cell contains both the channel and surrounding floodplain. In this case, most cell boundaries interface with other water-filled cells rather than channel banks, again making hydraulic radius less representative, and iii) Our model simulates both channelized flow and overland flow on hillslopes using the same equations. For overland flow, the concept of hydraulic radius is not applicable due to the absence of channel banks.

The hydraulic radius approach is most appropriate when cell size closely matches channel width, where a significant proportion of the flow surface contacts both the river bed and banks. Since it is difficult to predict this specific case across an entire model domain, we set water depth as the default in shear stress calculations. Nevertheless, users can override this default by setting *use_hydraulics_radius_in_shear_stress = True* when instantiating the RiverBedDynamics component, which calculates shear stress using:

$$\tau_x = \rho g R_h S_{fx} \text{ or } \tau_y = \rho g R_h S_{fy} \qquad\qquad \text{Eq. 6}$$

where $R_h = A_w/P_w$ is the hydraulics radius, $A_w = h\Delta x$ is the wetted area, and $P_w = 2h + \Delta x$ is the wetted perimeter. For north-south links we have $A_w = h\Delta y$ and $P_w = 2h + \Delta y$.

While our component focuses specifically on sediment transport by flowing water, we acknowledge that other important hillslope processes, such as soil creep and mass wasting, require different treatment. These processes can be represented by

coupling RiverBedDynamics with other Landlab components, such as TransportLengthHillslopeDiffuser for soil diffusion and LandslideProbability for mass wasting. Future versions of the model may also incorporate an option to estimate sub-grid channel widths for cases where channels are significantly narrower than grid cells.

### 3.3 Bed surface properties and sediment fluxes calculations

Prior to calculating sediment fluxes, RiverBedDynamics determines the bed properties required for the bed load transport equations. During instantiation, bed grain size distributions (GSD) are specified at nodes, which allows them to vary spatially. Grain sizes, defined as percentage passing, can range from fine sand to large boulders. Cohesive sediments are not supported by our component. For any sediment transport model used, it is mandatory to define the grain sizes at 0% and 100% of the distribution. This ensures uniformity in input format across different bed load transport equations.

Once the GSD is specified the model calculates the following parameters at nodes: sand fraction ($F_{sand}$), $D_{50}$, the geometric mean size ($D_{sg}$), and the geometric standard deviation ($\sigma_g$). These last two variables are defined following the method of Parker (1990). After bed properties are defined, they are mapped into the links assuming that the connecting nodes have equal weights. The selected bed load equation defines whether these bed surface properties are updated each time step or remain constant throughout the simulation (see below).

Six different sediment transport equations are available in our component. These equations are described in detail in the original articles (Fernandez Luque and Van Beek, 1976; Huang, 2010; Meyer-Peter and Müller, 1948; Parker, 1990; Wilcock and Crowe, 2003; Wong and Parker, 2006) and have been used extensively in sediment transport studies (e.g., Barry et al., 2004; Schneider et al., 2015; Yager et al., 2007). Therefore, only the aspects related to their implementation are described here.

The first group of equations include those by Meyer-Peter & Müller (1948), Fernandez Luque & Van Beek (1976), Wong & Parker (2006), and Huang (2010). We collectively refer to these as 'Meyer-Peter & Müller style equations.' They have the form:

$$q_b^* = \alpha(\tau^* - \tau_c^*)^\beta \qquad\qquad \text{Eq. 7}$$

where the coefficient $\alpha$, the exponent $\beta$, and $\tau_c^*$ are parameters specific to the selected equation. These equations are only valid when $\tau^* > \tau_c^*$ or else $q_b^* = 0$. When selecting these equations, the grain size distribution of the bed remains constant during the entire simulation. The dimensionless shear stress is calculated as $\tau^* = \tau/(\rho R g D_{50})$, where $R = (\rho_s - \rho)/\rho$, and $\rho_s$ and $\rho$ are the densities of the sediment and water, respectively. For simplicity, the $x$ and $y$ subscripts from Eq. 5 are omitted in this and subsequent uses of $\tau$.

The effect of bed slope on critical shear stress (Lamb et al., 2008; Mao et al., 2008; Mueller et al., 2005; Smith et al., 2023; Yager et al., 2012) in relatively steep slopes (larger than 3%) can be empirically included in Meyer-Peter & Müller style equations by setting the option *variable_critical_shear_stress = True* during instantiation. When activated, Mueller et al., (2005b) equation is used to calculate $\tau_c^*$ :

$$\tau_c^* = 2.18\, S_b + 0.021 \qquad \text{Eq. 8}$$

where $S_b$ is the topographic gradient defined as $\partial\eta/\partial x$ and $\partial\eta/\partial y$ for the $x$ and $y$ directions, respectively. While this optional setting simplifies the complex processes governing critical shear stress variation with slope, it serves as a practical tool for analyzing the model's response to $\tau_c^*$ in terrain slopes exceeding 3%. We acknowledge that this approach is a generalization and incorporating the full mechanics of these processes is outside the current scope of RiverBedDynamics.

Another option is the surface-based bed load transport equation of Parker (1990) that includes the effects of sediment mixtures in gravel-bedded rivers but does not include sand size material. In this case, if sand is present in the GSD the component will automatically remove it and renormalize the GSD curves to adjust for the change. The shear stress is here normalized using $D_{sg}$ instead of $D_{50}$ as follows:

$$\tau_{sg}^* = \tau/\left(\rho R g D_{sg}\right) \qquad \text{Eq. 9}$$

The dimensionless measure of shear stress is:

$$\phi_{sg0} = \frac{\tau_{sg}^*}{\tau_{rsg0}^*} \qquad \text{Eq. 10}$$

where $\tau_{rsg0}^* = 0.0386$ is the reference Shields stress. To account for the effects of sediment mixtures a hiding function is used:

$$\phi_i = \omega\phi_{sg0}\left(\frac{D_i}{D_{sg}}\right)^{-0.0951} \qquad \text{Eq. 11}$$

where the subscript $i$ denotes the $i^{\text{th}}$ grain-size class. The function $\omega$ is:

$$\omega = 1 + \frac{\sigma_g}{\sigma_0(\phi_{sg0})}\left[\omega_0(\phi_{sg0}) - 1\right] \qquad \text{Eq. 12}$$

where $\sigma_0(\phi_{sg0})$ and $\omega_0(\phi_{sg0})$ are functions that are calculated automatically within the component. The dimensionless transport rate for each $i^{\text{th}}$ size class is defined as:

$$W_i^* = 0.00218\, G(\phi_i) \qquad \text{Eq. 13}$$

and the function $G(\phi_i)$, the normalized dimensionless gravel bedload transport rate, is:

$$G(\phi_i) = \begin{cases} 5474\left(1 - \dfrac{0.853}{\phi_i}\right)^{4.5} & ,\phi_i > 1.59 \\ exp[14.2(\phi_i - 1) - 9.28(\phi_i - 1)^2] & ,1 \le \phi_i \le 1.59 \\ \phi_i^{14.2} & ,\phi_i < 1 \end{cases} \qquad \text{Eq. 14}$$

To obtain the fraction of bed load in each i[th] size class ($p_i$) we used:

$$p_i = \frac{F_i G(\phi_i)}{\sum_{i=1}^{N} G(\phi_i) F_i}$$

Eq. 15

where $F_i$ is the volume fraction in the bed of the i[th] grain-size class and $N$ is the number of grain size classes with characteristic diameters $D_i$.

The last bed load transport equation included in our component is Wilcock & Crowe (2003). Similar to Parker (1990) this model can handle sediment mixtures. However, in this case the effects of sand content ($F_{sand}$) are explicitly included in the

reference Shields stress, which is defined as:

$$\tau_{rsg0}^* = 0.021 + 0.015 exp(-20 F_{sand})$$

Eq. 16

The dimensionless measure of shear stress is $\phi_{sg0} = \tau_{sg}^* / \tau_{rsg0}^*$ and the hiding function is expressed as:

$$\phi_i = \phi_{sg0} \left( \frac{D_i}{D_{sg}} \right)^{-b}$$

Eq. 17

where the exponent is:

$$b = \frac{0.67}{1 + exp\left(1.5 - \frac{D_i}{D_{sg}}\right)}$$

Eq. 18

The dimensionless transport rate for each i[th] size class ($W_i^*$) is:

$$W_i^* = \begin{cases} 0.002\phi_i^{7.5} & , \phi_i < 1.35 \\ 14\left(1 - \frac{0.894}{\phi_i^{1.5}}\right)^{4.5} & , \phi_i \geq 1.35 \end{cases}$$

Eq. 19

To obtain the fraction of bed load in each i[th] size class ($p_i$) we used:

$$p_i = \frac{F_i W_i^*}{\sum_{i=1}^{N} W_i^* F_i}$$

Eq. 20

The volumetric bed load transport rate per unit width for each grain size when using Parker (1990) or Wilcock & Crowe (2003) is calculated using:

$$q_{bi} = \frac{(\tau/\rho)^{3/2} F_i W_i^*}{Rg}$$

Eq. 21

Given that we are working in a 2D structured grid we can assign directionality to $q_b$ depending on the link in which it is being calculated, $q_{b,x}$ for east–west and or $q_{b,y}$ for north-south links, by multiplying Eq. 21 by the sign of $\tau$. The total bed load transport rate per unit width $q_b$ is defined as the sum of the bed load transport rates of each grain size $q_{bi}$.

### 3.4 Sediment mass conservation and bed properties update

Once the sediment fluxes and bed load GSD at each link are calculated, it is possible to conduct a mass balance at nodes and determine changes in bed surface elevation and bed GSD. Surface bed elevation changes are calculated by the *update_bed_elevation* routine within RiverBedDynamics using the Exner equation:

$$(1 - \lambda_p)\frac{\partial \eta}{\partial t} = -\left(\frac{\partial q_{b,x}}{\partial x} + \frac{\partial q_{b,y}}{\partial y}\right) \tag{Eq. 22}$$

where $\lambda_p$ is the bed porosity. The equation states that the change in bed elevation in time within a control volume, a cell in this case, is a function of the sediment fluxes crossing the faces of a cell (Figure 4).

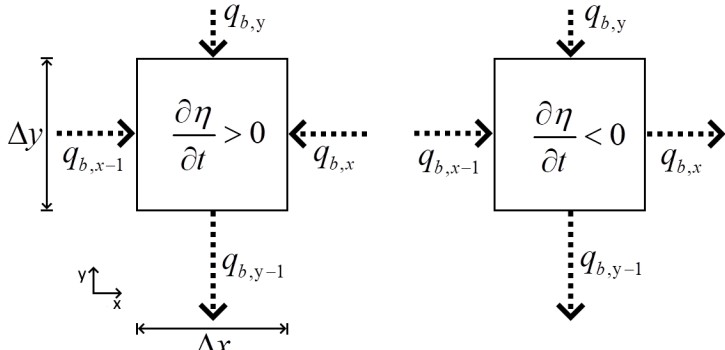

Figure 4: Examples of an increasing (left) and decreasing (right) bed surface elevation in time. Arrows indicate the direction of sediment fluxes across cell faces (arrow size adjusted for visibility), with flux directions determining the net bed load transport rate within a cell and consequently dictating erosion or deposition of sediment. In the left diagram, the sum of the three fluxes entering the cell is larger than those exiting the cells, therefore, there is a net accumulation of sediment, and the bed elevation increases. In the right diagram, fluxes in the $x$ direction are equal in magnitude and cancel each other out, whereas in the $y$ direction the flux leaving is larger than that entering, consequently the bed elevation will decrease.

We used an explicit method to approximate the solution of Eq. 22. The gradients in volumetric bed load transport rate per unit width in the $x$ and $y$ directions are:

$$\frac{\partial q_{b,x}}{\partial x} \approx \frac{q_{b,x} - q_{b,x-1}}{\Delta x} = \frac{\Delta q_{b,x}}{\Delta x} \quad \text{and} \quad \frac{\partial q_{b,y}}{\partial y} \approx \frac{q_{b,y} - q_{b,y-1}}{\Delta y} = \frac{\Delta q_{b,y}}{\Delta y} \tag{Eq. 23}$$

where the locations $x$, $x - 1$, $y$, and $y - 1$ are shown in Figure 4. The right-hand side of Eq. 22 can be expressed as:

$$\frac{\partial q_{b,x}}{\partial x} + \frac{\partial q_{b,y}}{\partial y} \approx \frac{\Delta q_{b,x}\Delta y + \Delta q_{b,y}\Delta x}{\Delta x \Delta y} \tag{Eq. 24}$$

here, $\Delta q_{b,x}\Delta y$ and $\Delta q_{b,y}\Delta x$ are the volumetric bed load transport rates in each direction, $\Delta q_{b,x}\Delta y + \Delta q_{b,y}\Delta x = \Delta Q_b$ is the net volumetric bed load transport rate, and $\Delta x \Delta y = A_{xy}$ is the area of a cell. Considering these definitions, the explicit solution to Eq. 22 is:

$$\eta^{t+1} = \eta^t - \Delta t \frac{\Delta Q_b}{\left(1 - \lambda_p\right) A_{xy}} \qquad \text{Eq. 25}$$

The choice of an explicit scheme for solving the Exner equation (Eq. 22 and 25) was motivated by its simplicity and ease of implementation, particularly when coupled with the OverlandFlow component. While OverlandFlow uses an implicit scheme for flow routing, which is advantageous for maintaining stability in hydrodynamic simulations, the explicit approach for sediment transport allows for straightforward integration of sediment fluxes and bed elevation updates at each time step.

However, the explicit scheme imposes stricter stability constraints compared to implicit methods. The time step ($\Delta t$) must satisfy the Courant-Friedrichs-Lewy (CFL) condition for sediment transport, which can be approximated as: $\Delta t \leq \left(1 - \lambda_p\right) \Delta x \Delta y / q_b$. This condition ensures that sediment fluxes do not exceed the capacity of the grid cells to accommodate changes in bed elevation within a single time step. In practice, the time step is further constrained by the need

to maintain numerical stability in the coupled system, particularly during rapid changes in bed elevation or flow conditions. Our testing shows that time steps of 1-5 seconds typically maintain stability for spatial resolutions of 25-100 meters in watershed applications, though these values should be adjusted based on specific flow and sediment transport conditions. We recommend that users start with small time steps (around 1 second) and gradually increase them while monitoring solution stability.

Boundary conditions for updating the bed surface elevation are only required at the links located immediately upstream of the watershed outlet. Two options can be specified, zero-gradient (default) or fixed-value. The zero-gradient condition allows the bed elevation at the outlet to evolve naturally based on upstream conditions, maintaining the same bed surface elevation slope as the immediately upstream reach. This approach permits sediment to be transported through the outlet

based on local transport capacity, rather than imposing an artificial constraint on sediment flux. The net sediment exchange at boundary cells is identical to that located upstream of the outlet, ensuring sediment flux out of the domain. The fixed-value condition, alternatively, maintains a constant bed elevation at the outlet, which can be useful when modeling systems with known base levels. However, it's important to note that sediment fluxes can be independently specified at any location in the domain, including inlet boundaries, outlet boundaries, and internal nodes and links. This flexible approach allows

users to implement various sediment supply scenarios and internal sediment sources or sinks.

In practice, the zero-gradient condition is implemented using user-provided methods for the watershed outlet boundary condition (e.g., *set_watershed_boundary_condition_outlet_id*, *set_watershed_boundary_condition*) to identify all connecting nodes and links upstream of the outlet. At the end of the bed elevation update routine, RiverBedDynamics adjusts the bed

elevation of outlet nodes to match the upstream slope, maintaining continuity in both elevation and sediment transport. This prevents artificial disruptions to sediment transport at the boundary while ensuring a physically consistent outlet condition.

When other types of boundary conditions are required, such as an elevation that changes in time following a given curve, it can be specified by setting individual nodes or links of the grid using Landlab boundary condition handling. Fixed-value conditions can be applied, not only to the boundaries, but also to internal nodes, such that they can remain unaltered throughout the whole simulation. This optional capability is accessed by editing the field *'bed_surf__elev_fix_node'*.

Sediment mass entering or leaving a cell can not only alter the bed surface elevation but also the bed GSD. We represent the evolution of the surface and substrate GSD by means of three layers (bed load, surface, and substrate; Figure 5). to capture key physical processes in gravel-bed rivers. This multi-layer approach is essential for accurately modeling: (1) the interaction between flow and the active surface layer where sediment exchange occurs, (2) the development of bed armoring and sorting patterns, and (3) the preservation of depositional history in the substrate, which influences future bed evolution. The bed load layer is the one defined by the bed material being transported close to the river bed and is calculated according to section 3.3. The surface layer, which is in direct contact with the flow at wet nodes, determines the bed surface elevation, measured from a specific datum point ($\eta$). It contains the active layer, characterized by a thickness defined as $L_a = 2D_{90}$ ($D_{90}$ is the 90th percentile of the surface GSD). The surface layer can exchange material with the bed load or substrate layer depending if the bed aggrades or degrades, respectively. We acknowledge that using a constant active layer depth is a simplification of the physical processes occurring in natural rivers, where the active layer depth can vary with flow strength and unsteady conditions. This assumption may affect the accuracy of predicted bed-substrate interactions, particularly during high-flow events when the actual active layer depth might be greater than our specified value. The constant active layer depth could potentially underestimate the exchange of material between the bed surface and substrate during high flows, limit the model's ability to capture deep scour events, and affect the predicted rate of bed armoring and sorting processes. However, this simplification was chosen to maintain computational efficiency and stability in watershed-scale applications. Future versions of the component could incorporate a variable active layer depth based on flow conditions, though this would require additional validation against field observations.To simplify the definition of the surface layer and facilitate the implementation of its updating algorithm, we adopted the definition of Toro-Escobar et al. (1996), which posits that the surface layer and the active layer are of equal thickness, $L_a$ (Figure 5). The substrate includes all the material below the surface layer. Its GSD is represented using $F_{s\,i}$, analogous to the way $F_i$ defines the surface GSD.

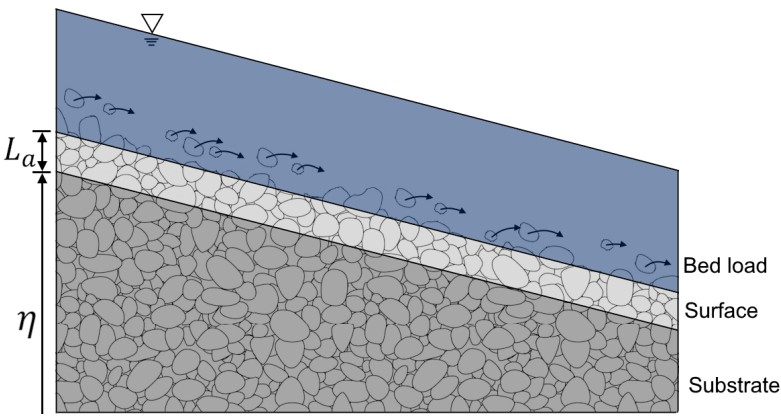

Figure 5: Schematic diagram of the model's three layers, used to represent the evolution of the surface and substrate grain size distribution.

To account for the dynamics of active layer grain sizes, we implemented the grain-size-specific form of the Exner equation as described by Parker (1991).

$$(1 - \lambda_p)\left( L_a \frac{\partial F_i}{\partial t} + \left( F_i - f_{Ii} \frac{\partial L_a}{\partial t} \right) \right) = -\frac{\partial(q_b p_i)}{\partial x} + f_{Ii} \frac{\partial q_b}{\partial x} - \frac{\partial(q_b p_i)}{\partial y} + f_{Ii} \frac{\partial q_b}{\partial y}$$

Eq. 26

where $f_{Ii}$ accounts for the interchange of sediment between the active layer and the substrate interface. This corresponds to the fraction of material in the i$^{th}$ grain size exchanged between these two layers. In our model we used the transfer function of Toro-Escobar et al. (1996):

$$f_{Ii} = \begin{cases} F_{s\,i} \,, \partial \eta / \partial t < 0 \\ 0.7 p_i + 0.3 F_i \,, \partial \eta / \partial t > 0 \end{cases}$$

Eq. 27

This equation states that when the bed degrades the active layer GSD is that of the substrate (Figure 6c). Conversely, when the bed aggrades, a mixture of surface and bedload material transfers to the substrate, thereby creating stratigraphy.


We solved Eq. 26 explicitly approximating the derivatives as:

$$\frac{\partial q_b}{\partial x} \approx \alpha \frac{q_{b,x} - q_{b,x-1}}{\Delta x} + (1 - \alpha) \frac{q_{b,x+1} - q_{b,x}}{\Delta x}$$

Eq. 28

$$\frac{\partial(q_b p_i)}{\partial x} \approx \alpha \frac{q_{b,x} p_{i,x} - q_{b,x-1} p_{i,x-1}}{\Delta x} + (1 - \alpha) \frac{q_{b,x+1} p_{i,x+1} - q_{b,x} p_{i,x}}{\Delta x}$$

Eq. 29

the $y$ direction has an equivalent discretization (just replacing $x$ for $y$). The coefficient $\alpha$ is used to switch from an upwind to central difference scheme. For stability purposes we opted for a default value of 1. The explicit solution to Eq. 26 is:

$$F_i^{t+1} = F_i^t - \frac{1}{L_a}(F_i - f_{Ii})(L_a^t - L_a^{t-1}) + \frac{\Delta t}{L_a(1 - \lambda_p)}\left( -\frac{\partial(q_b p_i)}{\partial x} + f_{Ii} \frac{\partial q_b}{\partial x} - \frac{\partial(q_b p_i)}{\partial y} + f_{Ii} \frac{\partial q_b}{\partial y} \right)$$

Eq. 30

For simplicity we dropped some $t$ superscripts, but all variables are evaluated at the current time step except for $L_a^{t-1}$.

Given that our model can predict temporal changes in bed surface elevation and GSD we implemented stratigraphy tracking capabilities, thus allowing a better representation of processes that are not purely erosional or depositional. Our model stores the current and past GSD and elevation of the surface and substrate across the whole watershed (Figure 6). At the beginning of a simulation the surface and substrate have, by default, the same GSD (Figure 6a). When deposition occurs at a given

location, RiverBedDynamics stores the elevation and GSD of the deposited sediment; this data is recorded at regular intervals determined by the variable *num_cycles_to_process_strat*. Once the accumulated deposited material at that location reaches the user-specifier vertical thickness (*bed_surf_new_layer_thick*, default value of 1 m, $L_s$ in Figure 6b) it is logged as a new layer of stratigraphy. The recorded GSD for this layer is a time-averaged value derived from all the sediment deposited over the last *bed_surf_new_layer_thick* meters, after which the process begins anew. In scenarios where a new

stratigraphic layer is being eroded, the model reads the stored data and adjusts $F_s$ and $F_{s\,i}$ based on the elevation of the layer being scoured (Figure 6d). When the bed surface is eroded below the initial bed surface elevation, the $F_s$ retains its original state from the beginning of the simulation (Figure 6c).

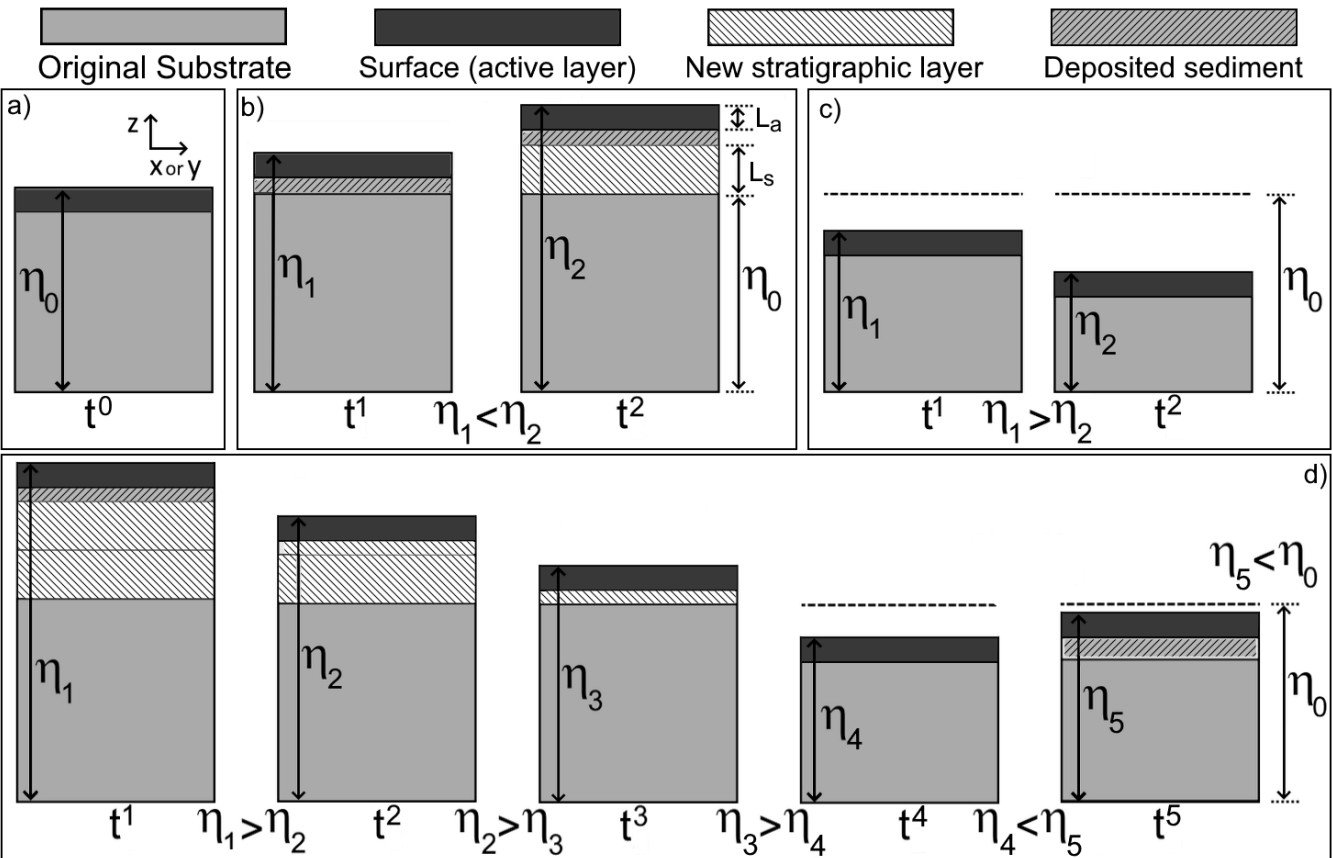

Figure 6: Graphical description of the model's algorithm for updating bed surface and substrate GSD. a) Initial bed surface elevation ($\eta_0$) at the start of the simulation ($t_0$). b) A pure depositional process. The bed surface elevation monotonically increases, new stratigraphic layers form once the deposited layer's thickness reaches $L_s$. The GSD of the deposited sediment is calculated using Eq. 26. c) A pure erosional case. Bed surface elevation monotonically decreases, the surface and substrate have the GSD specified at $t_0$. d) An alternating erosion/deposition case. The bed is first eroded below the initial elevation (at

$t_4$), following the erosion of newly deposited layers (at $t_1$ and $t_2$), and then experiences deposition again (at $t_5$). The local minimum bed surface elevation is updated to $\eta_4$ and the GSD at $\eta_5$ is calculated using Eq. 26.

## 4 Running a model using the Landlab framework

Some general characteristics of the Landlab modeling approach were described in Section 2.0. Therefore, in this section we focus only on describing specific details of the variables, default configurations, unit system, and capabilities of our model.

The component was designed to work exclusively using the International System of Units (SI). If imperial units are required they must be converted into SI before using them as input. Gravitational acceleration is constant and equal to 9.80665 m/s². During the instantiation of RiverBedDynamics the user can modify and/or define the variables or options listed in Table 1.

Table 1: List variables and options used in RiverBedDynamics during instantiation

| Variable or option | Default value | Units | Comment |
|---|---|---|---|
| $\rho_s$ | 2,650 | kg/m³ | |
| $\rho$ | 1,000 | kg/m³ | |
| bedload_equation | 'MPM' | - | See section 3.3 |
| $\lambda_p$ | 0.35 | - | |
| $\Delta t$ | 1 | s | |
| current_t | 0 | s | Used in case a simulation does not start at time 0 s |
| $\alpha$ | 1 | - | |
| outlet_boundary_condition | zeroGradient | - | Also available: fixedValue |
| surface_water__velocity_prev_time_link | Same as current time | m/s | Forces gradients in Eq. 4 to be zero. |
| variable_critical_shear_stress | False | - | |
| use_hydraulics_radius_in_shear_stress | False | - | |
| track_stratigraphy | False | - | |
| num_cycles_to_process_stra | 10 | - | |
| bed_surf_new_layer_thick | 1 | m | |

When using our component, like all other Landlab simulations, a driver file is required. This file is a procedure script containing a set of instructions to import libraries, instantiate classes, load data, run and loop through time, and finalize a

simulation. Once the elements have been initialized and are ready to loop in time, the two different basic routines that define our LEM are executed sequentially, first OverlandFlow then RiverBedDynamics (Figure 2). At every iteration within the time loop, OverlandFlow is executed and returns updated flow conditions (e.g., $q$ and $h$) across the domain and the $\Delta t$ required to predict changes in bed surface elevation and GSD (Eq. 25 and Eq. 26). Then, the first part of the RiverBedDynamics routine calculates and stores a series of hydraulics and sediment transport variables. When selecting a

bed load equation the following terminology is used: *MPM* for Meyer-Peter & Müller (1948), *FLvB* for Fernandez Luque & Van Beek, (1976), *Parker1990* for Parker (1990), *WilcockAndCrowe* for Wilcock & Crowe (2003), *Huang* for Huang (2010), or *WongAndParker* for Wong & Parker (2006). The default option is *MPM*. After all calculations are completed the second part of the RiverBedDynamics routine starts and uses the calculated bed load transport rates per unit width and bed load GSD to modify the bed elevation and GSD according to the equations described in Section 3.4.


The results of the calculations are stored as fields in the grid, but only the current time step is available for reading/writing, except for the velocity at the previous time step and stratigraphy properties. Therefore, when analyzing the changes of a given variable in time, the variable must be stored in a local file in a user-defined format that is specified in the driver file. The format in which RiverBedDynamics stores bed load, surface, and substrate GSD results may be difficult to interpret

because it was designed to be easily accessible by the component and not for user-readability. However, a postprocessing function called *format_gsd* is implemented and returns a panda DataFrame that contains the GSD for each node or link, depending on the input, in a user-friendly format.

## 5 Evaluation of RiverBedDynamics

### 5.1 Equilibrium bed surface slope in uniform flow conditions

To test the ability of our component for predicting changes in the bed surface elevation, we obtained an analytical solution for an idealized channel with uniform flow conditions. In this case, a given bed load transport rate is imposed at the upstream boundary such that the bed surface slope must adjust until the channel reaches a stable condition. We combined Manning's equation to include uniform flow conditions and Eq. 7 to estimate bed load transport rate within the channel. By expanding Eq. 7 we can solve for the bed slope required to transport an imposed bed load rate ($q_b$ in this case):

$$S = \frac{RD_{50}}{h}\left(\left(\frac{q_b}{\alpha\sqrt{RgD_{50}}D_{50}}\right)^{1/\beta} + \tau_c^*\right)$$

Eq. 31

The equilibrium slope ($S$) is a function of $h$ which in turn depends on the flow discharge ($Q$) and channel properties, in this case $n$ and channel width ($b$). Once the equilibrium state has been reached $h$ can be estimated using:

$$h = \left( \left( \frac{Q\,n}{b} \right) \left( \frac{1}{\sqrt{S}} \right) \right)^{3/5} \qquad \text{Eq. 32}$$

which is a form of Manning's flow equation considering a rectangular channel and shallow flows such that $b \gg h$, therefore, $R_h \approx h$. Combining Eq. 31 and Eq. 32 a solution for $S$ can be found.

$$S = \left( \frac{R D_{50}}{\left( \frac{Q\,n}{b} \right)^{3/5}} \left( \left( \frac{q_b}{\alpha \sqrt{R g D_{50}} D_{50}} \right)^{1/\beta} + \tau_c^* \right) \right)^{10/7} \qquad \text{Eq. 33}$$

Note that Eq. 33 is valid only under uniform flow conditions and may perform poorly at intermediate bed states (i.e., when the bed is adjusting) because the flow is not uniform locally. This form of the analytical solution is convenient when testing our component because, in terms of hydraulic variables, it only depends on $Q$, which can be specified as a boundary condition or by using a rainfall intensity that generates the target $Q$.

We conducted two tests to evaluate the response of our component. Both cases started with the same initial bed configuration but differed in the imposed upstream sediment supply rate. In general terms, they consisted of a 1500 m long, straight channel with an initial bed surface slope of 0.015 m/m, a fixed elevation at the outlet at 0 m, and a surface roughness of $n = 0.03874$. Flow discharge was constant $Q = 100$ m³/s and was specified by using a rainfall intensity of 0.01 m/s acting over a single cell of 100 m per side ($\Delta x = \Delta y$) located at the upstream boundary. This case represents essentially a one-dimensional flow scenario in which the grid is composed of uniformly sized cells, and the channel has a width of one cell. The digital elevation model (DEM) employed was 19 rows by 3 columns, with 2 of the columns serving as boundaries. The bed surface GSD was uniform with a grain size of 50 mm and the bed load transport equation was that of Meyer-Peter & Müller (1948). In OverlandFlow we specified $h\_init$, the initial water depth in all cells, as 1 mm, the time step was limited to a maximum of 5 s, and all other variables were left as their default value. The time limitation was imposed due to the rapid changes in bed elevation; while OverlandFlow alone could accommodate larger time steps based on the water flow Courant number, the coupled system required smaller time steps to maintain stability and prevent simulation crashes. This stability constraint arises from two sources. First, the explicit solution of the Exner equation imposes a theoretical stability criterion of $\Delta t \leq (1 - \lambda_p) \Delta x \Delta y / q_b$. Second, the coupling between flow and bed evolution introduces additional stability requirements, particularly during rapid morphological changes when there is strong feedback between bed elevation changes and flow conditions. In practice, we found that time steps satisfying the theoretical criterion may still need to be reduced by a factor of 2-5 to maintain stability in the coupled system, especially in areas of high sediment transport or rapid bed evolution. For the conditions in our test case (spatial resolution of 25-100 m, typical bed load transport rates), this resulted in stable time steps of 1-5 seconds. Users implementing different conditions should start with conservative time steps and adjust based on stability monitoring.

The modeled scenarios were a purely aggradation case in which $q_b = 0.0087$ m²/s and a purely degradation case where $q_b = 0.0012$ m²/s. We ran each case for 120 days of constant, steady flow, and compared the predicted and analytical bed slopes at the end of the simulation. We chose 120 days because at this time, the rate at which the bed elevations were changing were relatively small, $9 \cdot 10^{-5}$ and $-4 \cdot 10^{-4}$ m/day in the aggradation and degradation cases, respectively. We considered these rates small enough to be representative of an equilibrium condition.


In the aggradation case our LEM predicted an $S$ equal to 0.0251 whereas the analytical solution of Eq. 33 was 0.025 (percentage error of 0.32 %). The degradation case had an $S$ of 0.0101 and 0.010 by the LEM and analytical solution, respectively (1 % error). Locally, the major differences between the LEM-predicted and analytically solved bed elevations were in the upstream region, near where the sediment supply was imposed. During the final time step, the maximum local

percent difference in volume of deposited sediment was 0.09% (corresponding to an elevation change of 0.014 m) in the aggradation scenario and 0.68% (corresponding to an elevation change of 0.05 m) in the degradation scenario (Figure 7). The small percentage error and the general trend of the local surface elevation with streamwise distance (Figure 7) suggests that our component can accurately predict changes in bed elevation.

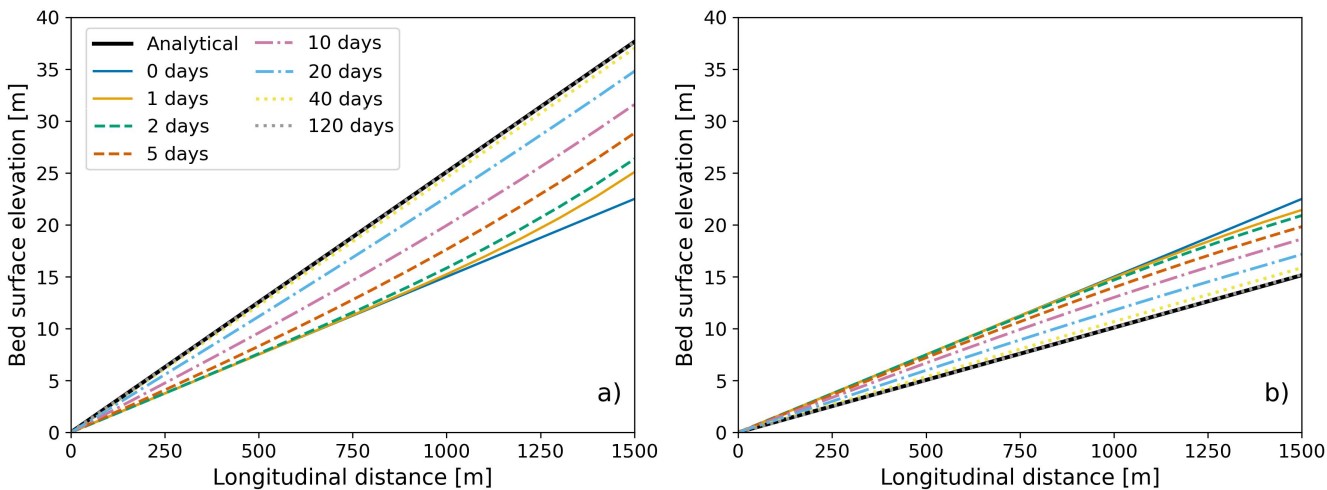


Figure 7: Changes in bed surface elevation for a case of a) pure aggradation and b) pure degradation. The analytical solution corresponds to the equilibrium slope given by Eq. 33. The small differences in bed elevation after 40 and 120 days indicate that the systems achieved an equilibrium state.

We analyzed the sensitivity of our results to the mesh size in the pure aggradation case by comparing the bed elevation after

120 days of simulation using meshes with half (50 m) and a quarter (25 m) of the size of the beforementioned case (Figure 8). By the end of each run, the average slope was 0.0251 in all cases. The percentage error in slope compared to the analytical solution was 0.29, 0.31, and 0.32 % for the 25, 50, and 100 m grid resolution, respectively. Other than the mesh

size, the configuration was identical in all simulations except for the maximum time step, which was 5 s for the 100 and 50 m cases and 2.5 s for the 25 m run.

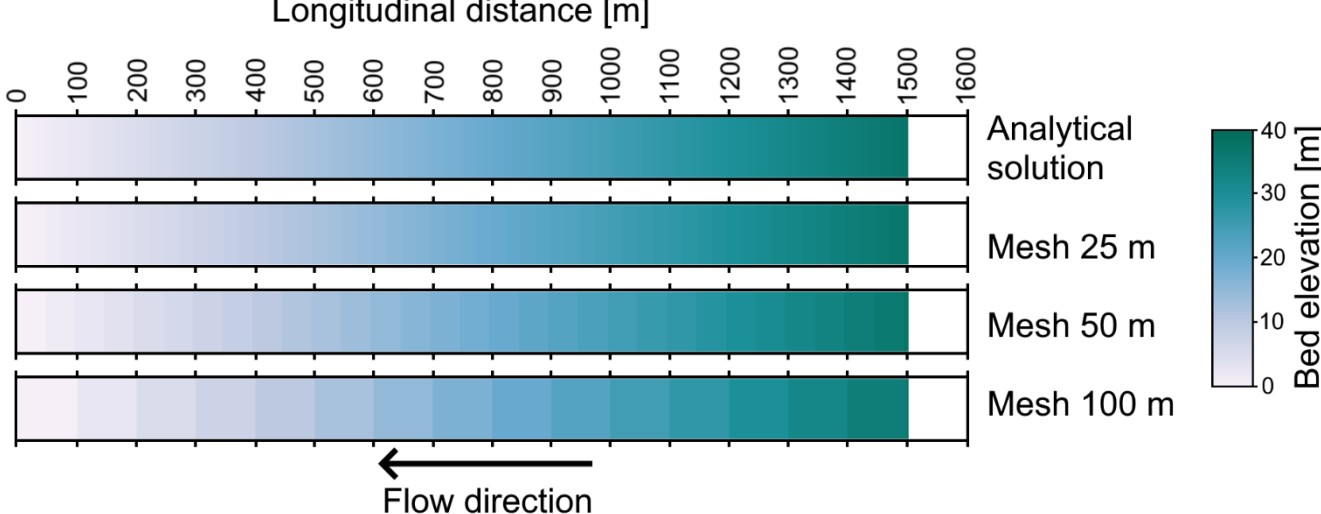


Figure 8: Sensitivity of the predicted bed elevations to the grid resolution after 120 days of simulation. Three different meshes were used and compared to the analytical solution.

### 5.3 Comparing bed load transport models predictions

We checked the predictions of bed surface elevation and local GSD for all bed load transport models included in our
component using a test similar to the one described in the previous section. In this case, we used a 1500 m long, straight channel with an initial bed surface slope of 0.015 m/m, a fixed elevation of 0 m at the channel outlet, a surface roughness of $n = 0.0275$, and a flow discharge $Q = 100$ m$^3$/s that was generated by a rainfall intensity of 0.02 m/s acting over two cells of 50 m per side ($\Delta x = \Delta y$) located at the upstream boundary. Similar to the previous case, this configuration models a one-dimensional flow setup, utilizing a grid of uniformly sized cells with a channel width of two cells. The initial bed surface
GSD had a $D_{50}$ of 32 mm and $D_{sg}$ of 28.84 mm including grains ranging from 2 to 256 mm (Figure 9 d). The initial water depth ($h\_init$) at all cells was 1 mm, and the time step was fixed and equal to 5 s. Similar to the previous test case, this time step constraint was necessary to account for the rapid bed elevation changes. Although OverlandFlow independently allows for larger time steps based on the Courant number, the coupled model demanded shorter intervals to ensure numerical stability. All other variables had their default value. The upstream sediment supply was $q_b = 0.0075$ m$^2$/s with the same GSD
as the initial bed surface. The total simulation time was 120 days for all the models we ran. We choose this test configuration because our LEM predictions using the bed load equations of Parker (1990) and Wilcock & Crowe (2003) can be verified using the algorithm developed and implemented by Parker (2004) RTe-bookAgDegNormGravMixPW.xls (Sediment transport morphodynamics with applications to rivers and turbidity currents, http://hydrolab.illinois.edu/people/parkerg/morphodynamics_e-book.htm). For the Parker (1990) equations, we considered

two scenarios: one where the GSD remains constant, and another where the surface and substrate GSDs are updated in line with Eq. 30. These scenarios are referred to as 'Parker' and 'Parker stratigraphy update' in Figure 9. Additionally, an analytical solution for the equations of Meyer-Peter & Müller (1948), Fernandez Luque & Van Beek (1976), Wong & Parker (2006), and Huang (2010) is available using Eq. 33.

We compared the predicted channel longitudinal profiles between all bed load transport models at different simulation times (Figure 9 a and b). After 10 days, the equations of Parker (1990) and Wilcock & Crowe (2003) predicted a more concave-upward- longitudinal profile and a higher elevation at the upstream boundary compared to the models that do not account for the whole GSD (Figure 9 a). The models of Meyer-Peter & Müller style equations had a more uniform slope along the channel profile. Comparing our LEM predictions with those from the RTe-bookAgDegNormGravMixPW.xls (hereinafter,

Parker-ebook), we observed good agreement in the predicted bed elevations along the channel. For Parker (1990) and Wilcock & Crowe (2003), the average errors in elevation were around 0.1%, with a maximum local difference in bed elevation of less than 1.5%. This corresponds to an elevation difference of 0.565 m at the upstream boundary for the Parker (1990) model that uses stratigraphy update. There was no analytical solution after 10 days for any model because the equilibrium condition had not been reached yet.


    After 120 days, all models were considered to be in relatively stable conditions, as indicated by the rate of elevation change at the upstream boundary node. The maximum elevation change was 22 mm/day for Wilcock & Crowe (2003), followed by 14 mm/day for Wong & Parker (2006), and less than 10 mm/day for all other models. Considering the increase of 52 m over this period at the upstream end of the model for Wilcock & Crowe (2003), the rate of 22 mm/day can be seen as relatively

minor. Although the longitudinal profiles from all models showed a relatively uniform slope (Figure 9 b), local elevations varied. For instance, Wilcock & Crowe, (2003) predicted a final bed slope of 0.0495 m/m, almost twice as steep as the slopes predicted by Meyer-Peter & Müller (1948) at 0.0249 m/m and Fernandez Luque & Van Beek (1976) at 0.0311 m/m. Parker (1990), with stratigraphy updates, predicted an average bed slope of 0.0408 m/m.

The differences in predicted equilibrium slopes among the models reflect their distinct approaches to representing sediment transport processes. For our specific test conditions, the Meyer-Peter & Müller style equations (MPM, FLvB, and Wong & Parker), which predict the lowest slopes (0.0249-0.0394 m/m), assume uniform grain size and represent transport as a simple power function of excess shear stress. In contrast, Wilcock & Crowe's model predicts steeper slopes (0.0495 m/m) due to its distinct hiding function, which affects the relative mobility of different grain sizes. Parker's model (0.0408 m/m) yields

intermediate slopes as it uses a different hiding function formulation in its surface-based approach to grain size mixtures. While these differences highlight how the choice of transport equation can significantly impact morphological predictions, particularly in systems with diverse grain sizes, it is important to note that these results are specific to our test conditions and should not be generalized to other scenarios without careful consideration of local conditions and grain size distributions.

Users should select a parameterization based on their specific application: Meyer-Peter & Müller style equations are suitable
for uniform sediment cases, while Parker or Wilcock & Crowe models are more appropriate for mixed-size sediments, especially when grain sorting processes are important.

Similar to the observations after 10 days, the elevation predictions of RiverBedDynamics aligned well with those in Parker-ebook and the analytical solutions. In terms of average percentage error, all predictions were below 1.4%, with a maximum
local difference in bed elevation of less than 1.1%. This discrepancy corresponds to an elevation difference of approximately 15 cm. The elevation predicted by the Meyer-Peter & Müller style equations in the LEM closely matched those calculated using the equilibrium slope for the same equations (Eq. 33), with errors below 0.5%.

Based on the results of Parker (1990) with stratigraphy updates, we analyzed the local evolution of the surface $D_{sg}$ at
different times during the simulation. Initially, the bed at the most upstream node quickly adjusted (Figure 9 c, 1 day panel) with $D_{sg}$ increasing from 28.84 to 33.19 mm. This value remained almost constant until the end of the simulation, with a final $D_{sg}$ of 33.45 mm. In the first 9 days of simulation, the bed also experienced locations of fining, indicated by local $D_{sg}$ values lower than 28.84 mm. However, after 10 days and until the end of the simulation, the bed consistently had a $D_{sg}$ larger than 28.84 mm across all locations. The upstream portion of the channel initially coarsened compared to the original
bed grain size because the supply of upstream finer sediment was less than the transport capacity of this sized material. Such finer sediment was therefore eroded from the most upstream cells, supplying finer sediment downstream. This upstream erosion and preferential transport of the finer sediment led to the pattern of downstream fining in the beginning of the simulation (e.g., 1-10 days). However, this pattern was temporary. Given that the upstream supply of finer sediment at the simulation boundary did not replace the removed fine bed material at a sufficient rate, fine sediment was progressively
winnowed throughout the entire model domain through downstream transport and vertical sorting into the subsurface. As this winnowing process continued, the initial downstream fining pattern was gradually replaced by widespread surface armoring, where coarser grains became concentrated at the surface, resulting in systematic coarsening across the domain at later timesteps. This sequence illustrates the dynamic interaction between selective transport, sediment supply, and local hydraulic conditions during bed adjustment, showing how temporal changes in sediment availability can shift the dominant
grain sorting pattern. On day 60, $D_{sg}$ was nearly uniform throughout the domain, with 33.4 mm at $x = 0$ and average of 33.3 mm).

To verify the accuracy of our $D_{sg}$ predictions, we compared them with those of the Parker-ebook. Despite small local differences in $D_{sg}$ (maximum of 1.04 mm on day 5), the magnitudes and spatial distribution matched reasonably well (Figure
9 c). The observed differences, though minor, can be attributed to the way flow is calculated. In our LEM, we used the results of OverlandFlow, a 2D flow solver, that accounts for flow unsteadiness while in the Parker-ebook the flow is

predicted using simplified relations for hydraulic resistance and the normal flow (local equilibrium) approximation. It's important to note that our goal was not to replicate the Parker-ebook results exactly but to have an approximate comparison to generally validate our findings.


Using this same example, we further explored the stratigraphy tracking capabilities of RiverBedDynamics, focusing on the comparison of surface and substrate GSD, particularly $D_{sg}$. For simplicity, we selected a single location at $x = 1000$ m and analyzed it through time, thereby not investigating spatial GSD changes in this analysis. In our graphical comparison (Figure 9 d), only the topmost layer of the substrate was considered. With the default setting of *bed_surf_new_layer_thick* at 1 m, the

first new layer was created after 8.1 days. This layer had a GSD that was, on average, coarser than the initial GSD (Figure 9 d). Throughout the 120-day simulation, a total of 12 layers were created, with the final one added after 93.9 days. Notably, ten layers were added before 50 days, and seven of these within the first 30 days. This pattern indicates that most substrate GSD updates occurred during the first quarter of the simulation, a period when bed conditions differed significantly from those observed at equilibrium (Figure 9 d subplot). The relatively minor changes in grain size distribution observed in our

simulations (Figure 9d) are consistent with the specific conditions tested, where we specified an upstream sediment supply with a grain size distribution matching the initial bed material GSD. As demonstrated by (Lei et al., 2023), different sediment supply characteristics can drive more pronounced grain size sorting patterns. Future applications of RiverBedDynamics could explore such scenarios by varying both sediment supply rates and GSD, though this was beyond the scope of this model description paper.


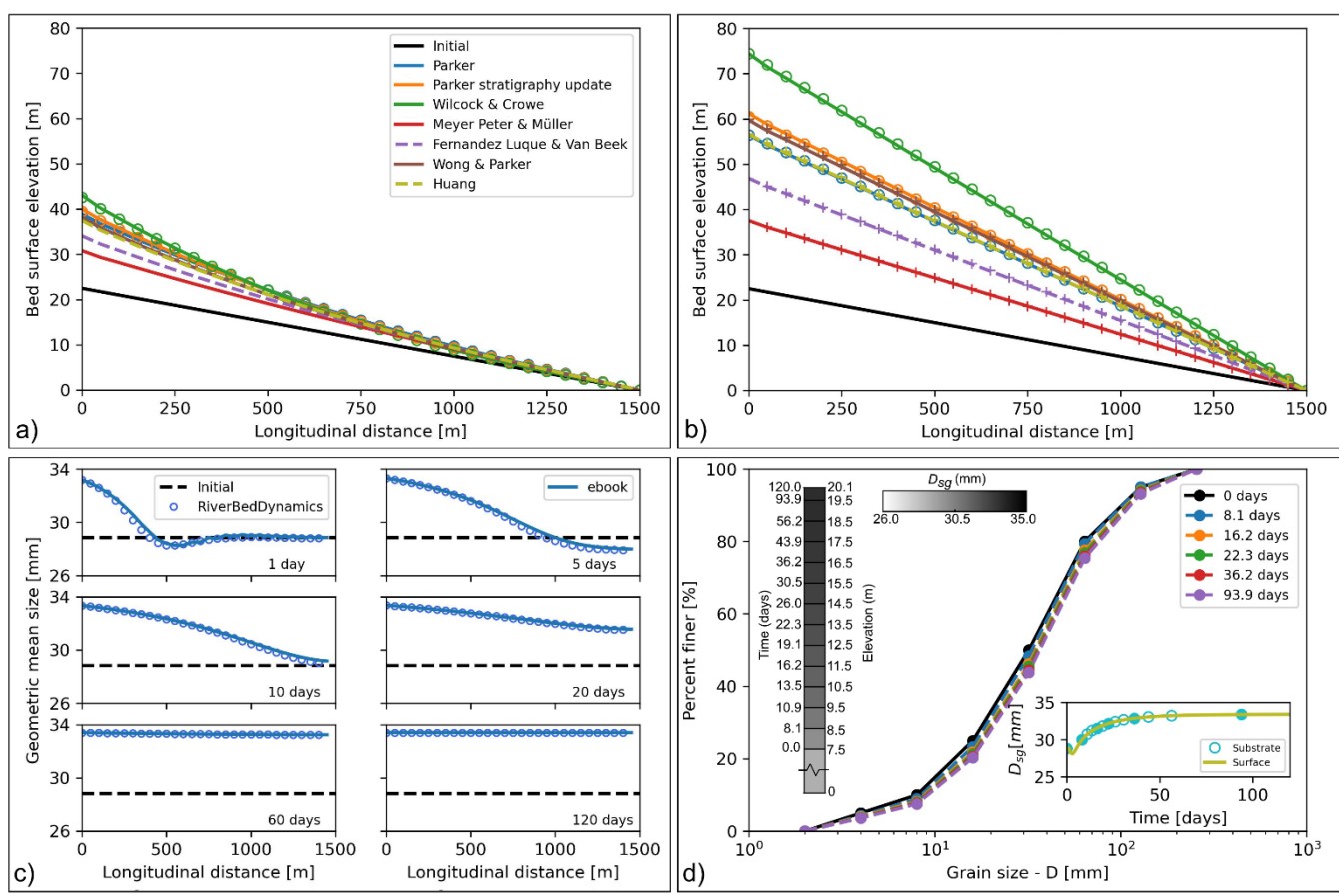

Figure 9: Evolution of bed surface elevation and local grain size distribution (GSD) for all bed load transport models implemented in RiverBedDynamics. a) Predicted longitudinal bed surface profiles after 10 days and b) after 120 days of simulation. Continuous and dashed lines represent RiverBedDynamics results, circles indicate solutions from the "ebook," and crosses denote analytical solutions. c) Changes in space and time of the bed surface geometric mean grain size $D_{sg}$. Initial values are repeated in each panel for reference. The "ebook" line represents the Parker-ebook solution, which is based on Parker (1990) equations. d) Substrate GSD evolution at $x = 1000$ m for various simulation times. Left of the GSD curve: stratigraphic profile showing layer formation times and elevations. Top values indicate final surface elevation (20.1 m) and simulation end time (120 days). Right subplot: Surface and substrate $D_{sg}$ temporal changes. The substrate in this context is the layer right below the surface as defined in Figure 5 and Toro-Escobar et al. (1996). Circle markers denote substrate GSD update times, with filled circles corresponding to times shown in the GSD curves.

## 5.4 Application to a large watershed – Effect of rainfall intensity on morphological changes

Our previous bed evolution tests predominantly focused on flow in a single channel and were restricted to pure erosion or deposition. To expand on this, we conducted a final test of our LEM in a more complex and larger watershed to analyze how flow discharge and bed surface elevation vary at different locations within the domain under different rainfall events. We

used a synthetic square watershed similar to that of Adams et al. (2017), covering an area of 36 km$^2$ with a resolution of 30 x 30 m per cell. The watershed elevations ranged from 0 m at the basin outlet to 25 m at the highest point (Figure 10 a).

For this watershed-scale application, we selected a 30 x 30 m grid spacing as it balances computational efficiency with the ability to capture key morphological features. While finer resolutions are possible, they become highly computationally demanding for large domains. This choice allows us to demonstrate the component's capabilities in capturing reach-scale morphodynamics while still enabling watershed-scale analysis. In a watershed of this size, grid cells are typically wider than natural channel widths, which influenced our approach to shear stress calculations. Specifically, we use water depth rather 730 than hydraulic radius for two main reasons. First, when a cell is wider than the natural channel, most cell boundaries interface with other water-filled cells rather than channel banks, making hydraulic radius less representative of actual flow conditions. Second, our model simulates both channelized and overland flow using the same equations. Since hydraulic radius relies on the presence of defined banks, it is not applicable in overland flow conditions, further justifying the use of water depth. Although using water depth simplifies channel geometry representation, it provides reasonable estimates of 735 reach-scale sediment transport across the watershed.

We considered two cases of temporal distributions. Both cases used spatially uniform rainfall with a total precipitation of 24 mm. We refer to these cases as i) Steady, where the rainfall intensity was 10 mm/hr lasting for two hours and 24 minutes (8640 s), and ii) Intermittent, where rainfall consisted of four cycles alternating between 60 mm/hr and 0 mm/hr, with each 740 of the two rainfall rates within a cycle lasting for 360 s (Figure 10 b). We quantified changes in flow discharge and bed surface elevation at three locations: Site 1 located at the watershed outlet, Site 2 located upstream of the outlet and at the confluence of the most downstream tributaries, and Site 3 located approximately at the center of the watershed (Figure 10 a).

We ran each model for 24 hours, setting Manning's n uniformly across the watershed with a value of 0.025, and using the 745 bed load transport equation of Meyer-Peter & Müller (1948) with a $D_{50}$ of 4 $mm$. All other variables during the instantiation of the components had default values. We simulated each rainfall case with and without activating RiverBedDynamics (4 cases in total) to analyze the effect of the selected temporal distribution of rainfall intensity on flow hydraulics (e.g., flow discharge) independent of morphodynamic changes that would also influence the hydraulics (without RiverBedDynamics) and to include the feedbacks between hydraulics and morphological changes (with RiverBedDynamics).
750

When running only OverlandFlow (i.e., RiverBedDynamics deactivated), the resulting hydrographs for both the steady and intermittent cases had relatively smooth shapes at the three sites (Figure 10b). Compared to the steady case, the intermittent scenario showed earlier and larger peak discharges at every site. For example, at Site 1 under steady conditions, the peak was 54.2 m$^3$/s arriving after 3.6 hours compared to 57.9 m$^3$/s at 2.6 hours under intermittent rainfall.
755

With RiverBedDynamics activated, the resulting hydrograph had a lower peak discharge compared to hydrographs run using fixed bed elevations. At the outlet, for the steady rainfall condition, the peak discharge was 40.7 $m^3$/s, a reduction of almost 25% compared to the case without bed evolution. For the intermittent case, the peak discharge was 42.1 $m^3$/s, a reduction of almost 27% compared to the case without bed evolution (Figure 10 b). At Site 2, the reductions in peak discharge for the steady and intermittent cases were about 19% when we included effects of bed evolution. At Site 3, the changes in hydrograph shape caused by including bed evolution were relatively small, with the discharge peak decreasing by less than 5% in both steady and intermittent scenarios.

Comparing hydrograph shapes, Sites 2 and 3 had a smooth shape, slightly skewed to the left, and with a single peak for both the steady and intermittent cases. Site 1, at the outlet, had similar characteristics to Sites 2 and 3 for the fixed bed elevation case. However, when the bed elevation varied in time, the shape of the hydrograph at Site 1 changed, featuring a double-peak.

RiverBedDynamics predicted alternating periods of erosion and deposition at Sites 1 and 2 for both rainfall cases. In the steady scenario, Site 1 initially eroded to -0.544 m from 0.023 m, before depositing to 0.256 m. Site 2 first deposited sediment, increasing elevation by 0.874 m, then eroded to 0.566 m. The intermittent case showed similar patterns, with Site 1 ranging between -0.526 m and 0.262 m, and Site 2 peaking at 0.922 m before reducing to 0.578 m. Site 3 showed negligible changes in both scenarios.

Most bed elevation changes occurred during the first 6 hours of simulation, coinciding with larger discharges and shear stresses. Throughout the watershed, scour and deposition patterns were observed primarily near confluences or areas with changes in local channel direction. The total area experiencing erosion or deposition larger than 1 cm after 24 hours was 5100 and 8730 $m^2$ for the steady and intermittent scenarios, respectively.

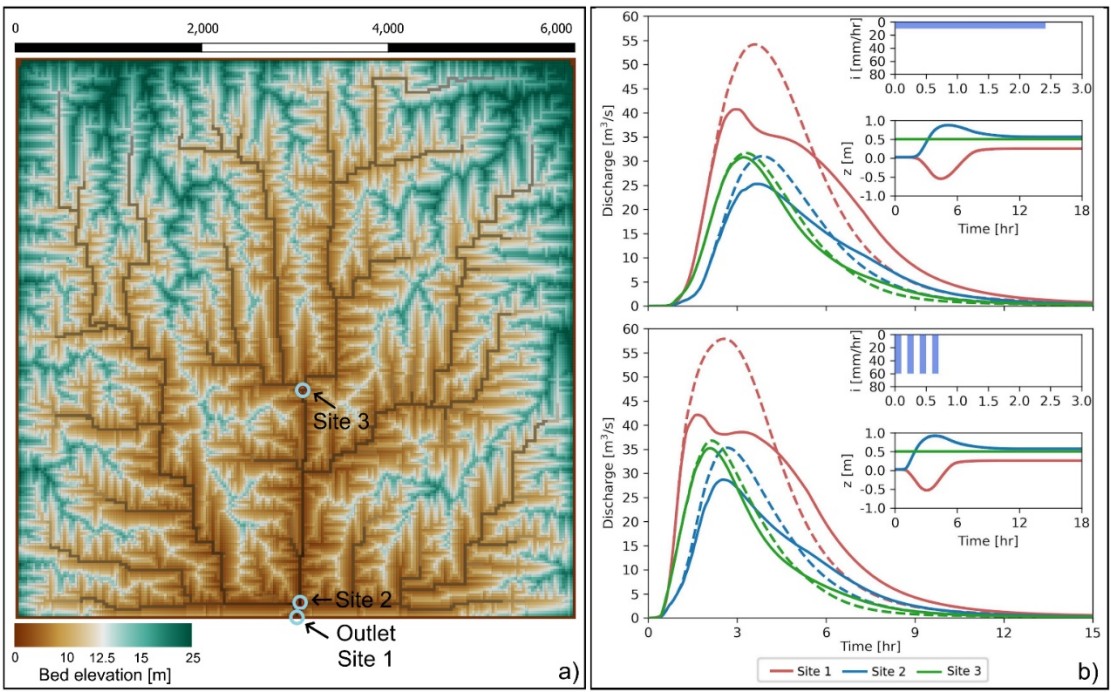


Figure 10: Discharge and bed surface elevation response to different rainfall intermittency scenarios. a) Synthetic square test basin. Three different sites, called Site 1, 2, and 3, were chosen to represent some of the spatial variability within the watershed. b) Hydrographs for steady (upper panel) and intermittent (lower panel) rainfall cases. Inset panels show hyetographs (top) and bed surface elevation (bottom) at the three sites. Dashed lines represent simulations with only OverlandFlow, while solid lines
include the effects of bed elevation changes predicted by RiverBedDynamics. Rainfall intensity in the hyetographs is plotted with the vertical axis inverted to better visualize the cascading relationship between rainfall input, discharge, and bed elevation response.

## 6 Discussion

The results presented in Section 4 demonstrate that RiverBedDynamics can be effectively coupled with a surface hydraulics
flow solver, specifically OverlandFlow in this study, to predict the evolution of bed surface properties at a watershed scale. This initial version allows us to simulate bed changes with varying degrees of complexity. For example, when utilizing the bed load transport equation of Meyer-Peter & Müller (1948), only changes in bed elevation can be considered. However, by selecting the equations of Parker (1990) or Wilcock & Crowe (2003), we can track changes in surface and substrate bed elevations as well as GSD over time. Thus, our LEM enables users to include or exclude certain processes and details
depending on their specific prediction requirements.

While our new component was developed using OverlandFlow, it can integrate with any flow solver available in Landlab due to the standardized component structure. For example, in very large watersheds, where local details are less crucial than regional changes, the KinwaveOverlandFlowModel could be used to reduce simulation time. Conversely, if small-scale

information is required, the OverlandFlow version of Adams et al. (2017) may suffice. It's worth noting that some
       assumptions that are included in the flow solver of Adams et al. (2017), such as negligible contributions from the advection
       term of the shallow water equations (Bates et al., 2010; de Almeida et al., 2012), may not be representative in a complex
       fluvial system. Consequently, a more comprehensive flow model may need to be developed to account for such processes.
       Regardless, the structure of RiverBedDynamics and other Landlab components facilitates easy model integration.


       We acknowledge that our model incorporates several simplifying assumptions that could potentially impact simulation
       accuracy in certain scenarios. First, our solution to the Exner equation, which predicts changes in bed surface elevation, is
       one of the simplest formulations when working in a 2D approach (Furbish et al., 2017a). While more generalized forms of
       sediment mass balances have been developed and applied (e.g., Juez et al., 2016; Paola & Voller, 2005; Parker et al., 2000),
we prioritized a computationally efficient implementation suitable for large watershed applications. In our formulation, we
       assumed that rectangular elements could define the alignment of a channel as well as general flow directions on hillslopes.
       However, this may not be representative of channels with significant curvature. Incorporating a curvature coefficient similar
       to that implemented by Van De Wiel et al. (2007) could lead to more accurate results, especially near river confluences.

Second, all our test cases involved channels without macro-roughness elements such as large boulders, vegetation, or any
       type of flow obstructions that can significantly alter the flow direction. While our component can theoretically be applied
       across a range of spatial scales (from meters to tens of meters), the computational demands of watershed-scale applications
       often necessitate coarser resolutions. In such cases, the effects of sub-grid scale features like macro-roughness elements
       would need to be parameterized rather than explicitly represented. Future development could incorporate methods to account
for these small-scale effects within coarser resolution simulations. Although we aimed to make RiverBedDynamics as
       versatile as possible, we have not yet evaluated its performance when subjected to sharp local gradients in shear stress
       induced by obstacles.

       Third, we implemented an optional slope-dependent critical shear stress equation (Mueller et al., 2005), which can be used in
the models of Meyer-Peter & Müller (1948) and Fernandez Luque & Van Beek, (1976). We recommend caution when using
       this option as it both overrides the original $\tau_c^*$ values and allows $\tau_c^*$ to vary in time as local bed slope change, which may lead
       to unexpected behavior. This capability was included based on preliminary model simulations where locations with steep
       elevation gradients, particularly riverbanks, eroded at a faster pace than expected, resulting in artificial channel widening.

Furthermore, certain sediment transport phenomena are not included in this first release. For example, RiverBedDynamics
       does not account for suspended sediment motion or its effects on bed evolution. Additionally, sharp unnatural angles within
       the river bed can occur because the effects of the angle of repose (sometimes called avalanche or sediment slide models)
       were not included in our results (Sanchez and Wu, 2011; Song et al., 2020). Finally, we did not incorporate the effects of

sediment or particle diffusion (Furbish et al., 2017b) that may smooth the bed profile, resulting in a more realistic

representation (compared to having large bed angles).

RiverBedDynamics is unique among Landlab components in its ability to predict sediment deposition using a fractional grain size formulation, making it particularly suited for modeling gravel bed rivers. Other components primarily focus on predicting bed surface elevation changes based on transport-limited or detachment-limited river assumptions. However, this

advanced capability comes at a computational cost: simulations using RiverBedDynamics can take up to 1.5 times longer compared to the DetachmentLtdErosion component (even when using the simplest configuration such as MPM, no stratigraphy tracking, and constant GSD). This increased runtime may constrain the total possible simulation time. Although there are no intrinsic limitations on simulation time in RiverBedDynamics, this mechanistic approach may be better suited for modeling relatively small-time scale processes.


The OverlandFlow-RiverBedDynamics approach in our LEM employs a a sequential solution strategy (Cao et al., 2002; Colombini and Stocchino, 2005). This means that the governing equations are solved separately and serially. Essentially, the flow is "paused" while the RiverBedDynamics component solves the Exner equation during a given time step (Figure 11 a and b). While the components interact through continuous feedback over multiple time steps, their equations are solved one

after another within each individual step. Consequently, the selected time-step must ensure relatively small bed elevation changes to maintain simulation stability. For scenarios involving flow, rainfall, and watershed conditions that generate dramatic elevation changes, an optional local correction can be used to preserve numerical stability and ensure mass conservation (Figure 11 c).

When bed elevation changes occur at a node, the initial calculation maintains the same water depth but shifts the water surface elevation by the same amount as the bed change. This can create unrealistic spatial jumps in water surface elevation between adjacent nodes that need to be smoothed while preserving discharge. The jumps occur because neighboring nodes still have their original water surface elevations. To address this, OverlandFlow runs for a few internal cycles, to redistribute around the nodes experiencing bed changes and their immediate neighbors (those sharing links). During these cycles,

simulation time does not advance, and the correction maintains mass conservation while achieving a smoother water surface profile. Once complete, the corrected water depths are mapped onto the grid, modifying local velocities while preserving discharge. This capability is configured in the driver file, with an example provided in the test case used in section 5.4.

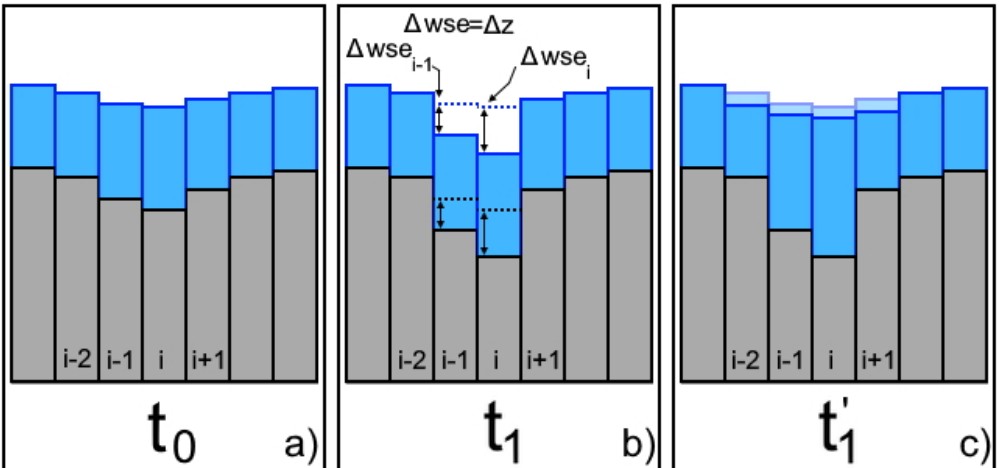

Figure 11: Illustration of the local water depth correction in RiverBedDynamics, demonstrated through an erosional case (similar principles apply to deposition). a) Initial bed and water depth condition ($t_0$). Grey rectangles represent individual bed nodes, identified by subscript $i$. Light blue rectangles depict water depth. b) Bed and water surface elevations change at nodes $i-1$ and $i$ by the end of time $t_1$, while water depth remains constant but creating unrealistic surface discontinuities. The bed erosion depth at a specific node equals the decrease in water surface elevation, $\Delta wse_{i-1} = \Delta z_{i-1}$ and $\Delta wse_i = \Delta z_i$, where $\Delta wse$ and $\Delta z$ represent the change in water surface and bed elevation, respectively. Dashed black and blue lines show the change in water surface and bed elevation at $t_0$. c) The local correction is applied to water surface elevation for all nodes sharing a link with those that experienced elevation changes (from $i-2$ to $i+1$), smoothing the water surface profile while preserving mass conservation. The time $t_1'$ denotes an internal cycle; simulation time does not progress during this correction. Transparent light blue areas above the water surface elevation indicate its position at time $t_0$.

# 7 Current capabilities and future enhancements

RiverBedDynamics represents a significant advancement in Landlab's modeling capabilities for river systems. As the first component to predict sediment deposition using a fractional grain size formulation, it is particularly suited for modeling gravel bed rivers. Unlike other components that focus primarily on bed surface elevation changes based on transport-limited or detachment-limited assumptions, RiverBedDynamics enables more complex simulations of bed evolution. It tracks changes in both surface and substrate bed elevations as well as grain size distribution over time, allowing users to model detailed and realistic scenarios of river bed dynamics at watershed scales.

The examples included with RiverBedDynamics demonstrate its versatility, yet they represent only a subset of the diverse scenarios that can be simulated within the Landlab environment. For instance, integrating RiverBedDynamics with other components like VegCA opens up new avenues for studying vegetation competition under non-steady sediment transport regimes. This integration capability highlights the component's potential for multidisciplinary research in fluvial geomorphology and ecology. RiverBedDynamics also enables researchers to investigate the impact of changing precipitation patterns on sediment transport and channel morphology, the role of grain size sorting in determining sediment delivery to reservoirs, the effectiveness of different river restoration designs that involve gravel augmentation, and the influence of

urbanization on channel stability through changes in both flow and sediment regimes. These applications demonstrate how the component's ability to simulate both erosion and deposition while tracking grain size evolution enables investigation of problems that were difficult to address with previous modeling approaches.

The model could also be used to understand how changes in climate influence bed evolution and GSD changes. Future applications could include coupling with bedrock erosion components to investigate how sediment cover and grain size distributions affect bedrock incision rates, though this would require modifications to incorporate a bedrock surface. Additionally, the component could be adapted to study sediment sorting and deposition patterns in alluvial fan environments, where changes in channel slope and width strongly influence grain size distribution and depositional processes.

While RiverBedDynamics already offers powerful modeling capabilities, there are exciting opportunities for future enhancements. One potential improvement would be implementing a time-varying Manning's roughness that responds to bed grain properties and water depth, such as the model proposed by Limerinos (1970). Additionally, future development could incorporate sub-grid parameterizations of bank erosion and channel migration processes. While our uniform grid framework cannot explicitly resolve channel widths, approaches similar to those developed by Van De Wiel et al. (2007) could inform how such processes might be represented through carefully designed parameterizations that account for grid-scale limitations. RiverBedDynamics could also benefit from adopting Landlab's unstructured grid handling system (ModelGrid). By extending the component's formulation to leverage Landlab's gridding library, the model could better represent irregular river geometries and heterogeneous topographic features in the surrounding landscape. To expand the component's applicability to longer timescales, we could implement a morphological acceleration factor (Morgan et al., 2020). This approach would allow for less frequent morphology calculations in slowly changing bed processes, extending the component's use to landscape evolution runs spanning millennia or longer while improving computational efficiency.

For applications in mountain river systems, particularly those with high gradient longitudinal slopes, implementing the equations developed by Schneider et al. (2015) and Yager et al. (2007, 2012) could provide more accurate bed load transport calculations. This addition would enable explicit inclusion of large roughness elements and sediment supply limited conditions common in steep streams. Another potential refinement is the inclusion of a critical shear stress that evolves with sediment transport rate, similar to the approach of Johnson (2016). These potential enhancements build upon the strong foundation of Landlab and in particular our proposed component RiverBedDynamics, expanding their already significant capabilities in modeling complex river systems across various spatial and temporal scales.

**8 Conclusion**

We presented the first version of RiverBedDynamics, a Landlab component designed and built to simulate 2D sediment transport and river bed evolution with a special focus on gravel-bedded rivers. Integrating RiverBedDynamics with a OverlandFlow has created a LEM capable of providing accurate and detailed predictions of bed surface evolution in terms of elevation and grain size distribution. This new LEM is physically based and solves fundamental governing equations such as the conservation of mass in RiverBedDynamics, and mass and momentum in OverlandFlow, enhancing its reliability for simulating unsteady processes. The new component is flexible enough for short- and long-term simulations depending on the

number of processes that can be included in each case. Our LEM's predictions were validated against analytical and previously reported solutions, demonstrating accurate representation of changes in bed surface elevation and grain size distribution. Both purely erosional and depositional cases were evaluated, with processes well captured in each scenario. Additionally, we employed a synthetic watershed to illustrate how the interaction between rainfall intensity distribution and sediment transport processes influences flow discharge and bed surface evolution across the domain.


While we have designed the first version of RiverBedDynamics to be as comprehensive as possible in representing sediment transport processes, there is potential for further enhancements and generalizations to expand its capabilities. Nonetheless, our LEM has demonstrated that the combination of OverlandFlow and RiverBedDynamics offers significant potential for simulating many typical scenarios encountered in practical river management situations and fundamental scientific research.

We anticipate that future developments will focus on improving the representation of bank erosion, channel migration, and local angle of repose effects.

**9 Code and data availability**

The source code for RiverBedDynamics is available in a public Zenodo repository at

https://doi.org/10.5281/zenodo.14159914. This repository contains the latest release version of the component, including all necessary files for running the model within the Landlab framework.

The example scripts and data used to create and run the test cases presented in this article are accessible in a separate Zenodo repository: https://doi.org/10.5281/zenodo.14159904. This repository includes all the necessary input files, parameters, and

scripts to reproduce the results discussed in this paper.

Both repositories are open-source and freely available for use, modification, and distribution under MIT License. We encourage users to refer to the README files in each repository for detailed instructions on installation, dependencies, and usage. For any questions regarding the code or data, please contact the corresponding author.


## 10 Author contribution

AM, SA, NG, and EY conceptualized the component and defined its requirements; AM developed the component code; SA performed alpha and beta testing; NG conducted code review and implemented improvements; AM developed the test cases with SA reviewing them; AM wrote the original manuscript draft; SA, NG, and EY reviewed and edited the manuscript.

## 11 Competing interests

The authors declare that they have no conflict of interest.

## 12 Acknowledgements

This work was supported by NSF award number 1918459 to Anderson and Gasparini. We are grateful for the insightful
conversations with Joel P. L. Johnson and Grace Guryan, which were instrumental in initiating this project. We extend our sincere appreciation to Eric Hutton and Mark Piper from CSDMS for their invaluable assistance during the development of this Landlab component.

## 13 Notation

Table 2 summarizes all variables and symbols used throughout the manuscript, organized by their physical meaning and
application within the model:

Table 2: Summary of variables used in RiverBedDynamics, organized by physical categories. Each variable is presented with its symbol, definition, and units (where applicable)

| Category: Geometric and Grid Variables | | Category: Flow Variables | |
|---|---|---|---|
| Variable | Definition and units | Variable | Definition and units |
| $\eta$ | bed surface elevation [m] | $h$ | water depth [m] |
| $\Delta x, \Delta y$ | grid cell dimensions in x and y directions [m] | $Q$ | flow discharge [m³/s] and |
| $A_{xy}$ | cell area [m²] | $q$ | flow discharge per unit width [m²/s] |
| $b$ | channel width [m] | $U$ | depth-averaged velocity [m/s] |
| $S_b$ | topographic gradient [-] | $u, v$ | velocity components in x and y directions [m/s] |
| $wse$ | water surface elevation [m] | $S_{fx}, S_{fy}$ | friction slopes in x and y directions [-] |

| Variable | Definition and units | Variable | Definition and units |
|---|---|---|---|
| $\Delta wse$ | change in water surface elevation [m] | $n$ | Manning's roughness coefficient [s/m1/3] |
|  |  | $A_w, P_w$ | wetted area [m²] and wetted perimeter [m] |
|  |  | $R_h$ | hydraulic radius [m] |

| Category: Sediment Properties | | Category: Transport Parameters | |
|---|---|---|---|
| Variable | Definition and units | Variable | Definition and units |
| $D_{50}$ | median grain size [m] | $\tau$ | shear stress [N/m²] |
| $D_{sg}$ | geometric mean size [m] | $\tau_x, \tau_y$ | shear stress components [N/m²] |
| $\sigma_g$ | geometric standard deviation [-] | $\tau^*$ | dimensionless shear stress [-] |
| $F_i$ | volume fraction in bed of ith grain size class [-] | $\tau_c^*$ | critical dimensionless shear stress [-] |
| $F_{si}$ | volume fraction in substrate of ith grain size class [-] | $q_b$ | volumetric bed load transport rate per unit width [m²/s] |
| $F_{sand}$ | sand fraction [-] | $q_b^*$ | dimensionless volumetric bed load transport rate [-] |
| $\lambda_p$ | bed porosity [-] | $q_{bi}$ | bedload transp. rate per unit width for i$^{th}$ size class [m²/s] |
| $\rho_s, \rho$ | sediment density [kg/m³] and water density [kg/m³] | $\Delta Q_b$ | net volumetric bed load transport rate [m³/s] |
| $R$ | submerged specific gravity [-] | $\alpha, \beta$ | bed load equation parameters [-] |

| Category: Stratigraphy Variables and others | | Category: Parker and Wilcock & Crowe Parameters | |
|---|---|---|---|
| $L_a$ | active layer thickness [m] | $\phi_{sg0}$ | dimensionless measure of shear stress [-] |
| $L_s$ | stratigraphic layer thickness [m] | $\tau_{rsg0}^*$ | reference Shields stress [-] |
| $f_{Ii}$ | transfer function between active layer and substrate interface [-] | $\phi_i$ | hiding function |
| $t$ | time [s] | $\omega$ | hiding function parameter [-] |
| $g$ | acceleration due to gravity [m/s²] | $W_i^*$ | dimensionless transport rate for ith size class [-] |
|  |  | $p_i$ | fraction of bed load in ith size class [-] |
|  |  | $G(\phi_i)$ | normalized dimensionless bedload transport rate [-] |

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
