# Peer review of "RiverBedDynamics v1.0: A Landlab component for computing twodimensional sediment transport and river bed evolution"

_EGUsphere, 2024_

## Author Response (AR1)

**Responses to Reviewer's Comments**
**RiverBedDynamics v1.0: A Landlab component for computing two-dimensional sediment transport and river bed evolution**
By: Angel D. Monsalve, Samuel R. Anderson, Nicole M. Gasparini, Elowyn M. Yager

**Reviewer #1**

We sincerely thank the reviewer for their thorough and constructive evaluation of our manuscript. Their suggestions have helped us identify areas where additional context and clarification would strengthen the manuscript. We especially appreciate the reviewer's attention to recent developments in the field, which has allowed us to better position our work within the current scientific literature. We have carefully addressed each comment and made corresponding revisions to improve the manuscript's clarity, completeness, and scientific rigor. Below, we provide point-by-point responses to all comments and detail the changes made to the manuscript.

**General comments**

1)  *The authors presented a simple approach to employ deposition process based on the mass conservation at each cell. Some previous works have also employed the sediment deposition using different frameworks such as size-dependent sediment deposition probability in a Lagrangian framework in CIDRE (Carretier et al. 2023, https://gmd.copernicus.org/articles/16/6741/2023/), and Eularian framework in River.lab (Davy et al, 2017 https://doi.org/10.1002/2016JF004156), and Minor et al. 2022 (https://doi.org/10.1029/2021JF006546). These previous works employing deposition process deserves a credit and need to be discussed how their sediment deposition framework complement or differs from this this work.*

R:  We appreciate the reviewer's suggestion to discuss previous works on sediment deposition modeling. We agree this will strengthen the manuscript's context and highlight our component's unique contributions. We have expanded our introduction section to include detailed discussion of these prior approaches:

At ~new line 45, we have added:

"For instance, CIDRE (Carretier et al., 2023) employs a Lagrangian framework with size-dependent sediment deposition probability, while River.lab (Davy et al., 2017) implements an Eulerian approach. More recently, Le Minor et al. (2022) further enhanced deposition modeling approaches. While these frameworks have successfully captured key aspects of sediment transport and deposition, RiverBedDynamics offers unique advantages through its seamless integration with the Landlab platform, allowing easy coupling with any flow solver, and its comprehensive tracking of temporal grain size distribution evolution"

2)  *The uniqueness of RiverBed Dynamics lies in its easy coupling with any flow solver, incorporation within the Landlab framework, and prediction of temporal changes in GSD (although GSD doesn't change much during the simulation period, Figure 9d). In another work, Lei et al (https://doi.org/10.1029/2023WR035983) studied the change in GSD in response to different sediment supply. The authors can try to alter the sediment supply rates*

*and supply GSD to study the sediment sorting (not necessary for this review as this is a model description paper, just a suggestion) or atleast discuss in the paper about the possible causes of minimal changes in the GSD.*

R: We appreciate the reviewer's observation about the minimal changes in grain size distribution during our simulations. We have addressed this point by adding a detailed explanation in Section 4.3, after the discussion of substrate GSD evolution:

" The relatively minor changes in grain size distribution observed in our simulations (Figure 9d) are consistent with the specific conditions tested, where we specified an upstream sediment supply with a grain size distribution matching the initial bed material GSD. As demonstrated by Lei et al. (2023), significant changes in sediment supply characteristics can drive more pronounced grain size sorting patterns. Future applications of RiverBedDynamics could explore such scenarios by varying both sediment supply rates and GSD, though this was beyond the scope of this model description paper."

This addition provides important context for the observed behavior in our simulations and acknowledges the potential for more dynamic GSD evolution under different conditions, as demonstrated in recent literature. While exploring various sediment supply scenarios would be valuable, we focused on demonstrating the basic functionality and validation of the RiverBedDynamics component in this initial presentation.

**Specific comments:**

1) *Line 125: key aspects can be mentioned as the bracketed term.*

R: Thank you for this suggestion. We have revised line 125 to explicitly state the key aspects that RiverBedDynamics can capture: "Through a series of tests and applications, we illustrate how RiverBedDynamics can capture key aspects (sediment transport dynamics, grain size evolution, erosion and deposition processes, and watershed-scale bed surface changes) of landscape evolution, particularly in systems dominated by gravel-bed rivers."

2) *Line 158-159: Rewrite the sentence*

R: We agree that this sentence needed revision for clarity. We have rewritten the text as follows: "Implementation of Eq. 1 requires calculating $\tau^*$ at each node and determining locations where $\tau^*$ exceeds $\tau c^*$ to compute $qb^*$. Using Landlab's structure and tools, Eq. 1 can be implemented as:"

3) *Figure 2: The fonts in Figure 2 are too small to read. The Figure with enlarged fonts in landscape format (rotating $90^O$) might improve the readability.*

R: We thank the reviewer for this formatting suggestion. We have modified Figure 2 as follows:

- Increased all font sizes for better readability
- Reorganized the layout vertically, placing the workflow boxes in a single column rather than side by side
- Reduced whitespace and adjusted box dimensions to maximize space utilization within the page width These changes significantly improve the figure's legibility while maintaining all the essential information about the model's workflow and coupling.

4) *Line 196-200: Can you please briefly mention what variables have been calculated at nodes and links specifically.*

R:  We appreciate this suggestion for improved clarity regarding the spatial discretization of variables. We have revised the text to explicitly state which variables are calculated at links versus nodes:

"During each time step of a simulation, OverlandFlow solves the 2D flow equations across all grid links determining the surface water discharge per unit width (q), flow velocity components (u and v, corresponding to the x and y directions, respectively) and water depth (h) at links. Subsequently, water depth at nodes is calculated based on mass conservation, factoring in all inflow and outflow at a given node. These spatial distributions of flow variables provide the foundation for subsequent sediment transport calculations."

5)  *Line 208-209: Why authors prefer central difference scheme as it calculates the velocity averaged over three links (x-1, x, x+1). Instead the velocity calculated between two links as . The velocity calculated between two links might be better representative of local channel hydraulics, (i.e., flow resistance). However, the form adopted might be better for numerical stability. Justify the choice in the manuscript.*

R:  We appreciate this thoughtful observation about our numerical scheme choice. We have added a brief explanation in the manuscript to justify our use of the central difference scheme. After Figure 3, we have added:

"While a two-point scheme, like upwind, provides better numerical stability in advection-dominated flows, we opted for the central difference scheme primarily for we opted for the central difference scheme primarily because it provides more accurate approximations of spatial gradients (with errors proportional to the square of the grid spacing) compared to first-order schemes like upwind differencing and its ability to capture multi-directional information propagation in our coupled system. This choice reflects a balance between computational accuracy and physical representation, particularly important when simulating complex watershed-scale processes with varying flow conditions. The scheme's higher-order accuracy and multi-directional nature are especially beneficial for resolving flow dynamics at channel confluences and during rapid flow variations, where information propagates in multiple directions simultaneously"

6)  *Line 238-247: Again, please clarify what parameters being calculated at nodes and links.*

R:  We thank the reviewer for this request for clarification. We have revised this section to explicitly state where each parameter is calculated and how they are mapped between nodes and links. The revised text now reads:

"Prior to calculating sediment fluxes, RiverBedDynamics determines the bed properties required for the bed load transport equations. During instantiation, bed grain size distributions (GSD) are specified at nodes, which allows them to vary spatially. Grain sizes, defined as percentage passing, can range from fine sand to large boulders. Cohesive sediments are not supported by our component. For any sediment transport model used, it is mandatory to define the grain sizes at 0% and 100% of the distribution. This ensures uniformity in input format across different bed load transport equations.

Once the GSD is specified, the model calculates the following parameters at nodes: sand fraction (Fsand), D50, the geometric mean size (Dsg), and the geometric standard deviation (σg). These last two variables are defined following the method of Parker (1990). After bed properties are defined, they are mapped into the links assuming that the connecting nodes have equal weights. The selected bed load equation defines whether these bed surface

properties are updated each time step or remain constant throughout the simulation (see below)."

7) *Equation 15: , Authors can clarify the role of $F_i$ in the numerator as transport of $i^{th}$ size can be estimated as transport of $i^{th}$ size divided by the transport of all size classes without $F_i$.*

R: We thank the reviewer for bringing this to our attention. There is a typographical error in Equation 15. The correct equation, following Parker (1990), should read:

pi = FiG(φi)/∑(FiG(φi))

where Fi appears both in the numerator and denominator. This form correctly represents the fractional transport rate as defined in Parker (1990, equation 29b). We have verified that this equation is correctly implemented in our code, and the error appears only in the manuscript text. We have corrected this typo in the revised version.

8) *Line 335-336: The active layer depth may change with the strength of flow (discharge, and depth), especially the unsteady flow. The assumption of constant active layer might intercept the interaction between bedload and the substrate layer, and might not be the better representative of the actual process in the field. The authors should justify the limitations of the approach and what implication it might have on the bed evolution and grain size distribution.*

R: We thank the reviewer for this important observation regarding the limitations of our constant active layer depth approach. We have added the following discussion to the manuscript after ~new line 352 to address these concerns:

"We acknowledge that using a constant active layer depth is a simplification of the physical processes occurring in natural rivers, where the active layer depth can vary with flow strength and unsteady conditions. This assumption may affect the accuracy of predicted bed-substrate interactions, particularly during high-flow events when the actual active layer depth might be greater than our specified value. The constant active layer depth could potentially underestimate the exchange of material between the bed surface and substrate during high flows, limit the model's ability to capture deep scour events, and affect the predicted rate of bed armoring and sorting processes. However, this simplification was chosen to maintain computational efficiency and stability in watershed-scale applications. Future versions of the component could incorporate a variable active layer depth based on flow conditions, though this would require additional validation against field observations."

9) *Line 360-370: Does the two stratigraphic layers may have different GSD depending on the time evolution of bedload? How does the stratigraphic deposition improve the process representation? What difference does it make with a single deposition layer instead of a deposition layer of a constant depth of 1 m. Depending on the discussion, it seems all the stratigraphic layers might possess merely the same GSD. Please clarify.*

R: We thank the reviewer for these thoughtful questions about our stratigraphic implementation. The implementation of multiple stratigraphic layers, rather than a single deposition layer, allows our model to preserve the temporal evolution of bed material properties, especially in the long term. Each new layer can potentially have a different GSD depending on the bedload composition at the time of deposition. The 1-meter thickness for new layers was chosen as a compromise between computational efficiency and stratigraphic resolution. However, this is a user-defined value (bed_surf_new_layer_thick in Table 1) and 1 m is the default value. A thinner layer would provide higher temporal resolution in depositional history but increase

computational overhead, while a thicker layer would reduce temporal resolution but improve computational efficiency.

The reviewer correctly observes that in our test case, the stratigraphic layers show similar GSDs. This similarity is intentional for this particular case, as we used constant sediment supply GSD specifically to validate our model against Parker's e-book solution, which provides a well-established benchmark for our implementation. However, the model is capable of handling varying GSDs between layers when sediment supply conditions change. This capability would be particularly relevant when simulating scenarios with time-varying sediment inputs or in watersheds with diverse sediment sources, though such applications were beyond the scope of this initial model description paper.

We believe the current manuscript text adequately describes the technical implementation, while this response provides the additional context for our validation approach.

10) *Table 1: Option surface_water__velocity_prev_time_link was not explained or m mentioned anywhere else in the paper.*

R:  We thank the reviewer for bringing this to our attention. We have added an explanation of this parameter in Section 3.1, after Equation 4, where we discuss velocity time gradients:

"… where $\Delta t$ is the time step and the superscripts t and t-1 indicate the current and previous time steps. The velocity at the previous time step (t-1) is stored in the field surface_water__velocity_prev_time_link and is updated every time step. At the beginning of the simulation, this gradient can be forced to zero or initialized with a known velocity value if available as an initial condition."

We believe that this addition provides the necessary context for the parameter and explains its role in the unsteady friction slope calculations mentioned anywhere else in the paper.

11) *Figure 9d: Its difficult to see the differences in GSD at different times. The authors can try to reduce the linewidth and reduce the frequency of plots and see if it improves the readability of the Figure (i.e. plotting only o, 8.1,16.2, and 93.9 days). Also Figures 9c and 9d shows minimal changes in GSD after the 60 days. Does this relate to the sediment sorting/armouring being developed after 60 days? The authors can add another subplots showing the vertical stratigraphic distribution of geometric mean size (at the same location x=1000m) and the GSD at different stratigraphic layers. Or Does it relates to continuous sediment supply with same GSD as that of bed. Also the authors can discuss the possible implications of altering the sediment supply rates and GSD on bed evolution.*

R: We appreciate the reviewer's suggestions regarding Figure 9d. While reducing the number of plotted time steps might increase visual clarity, we believe the current presentation has value in showing the continuous temporal evolution of the GSD, particularly the subtle changes that occur throughout the simulation. The overlap of the GSD curves after 60 days is actually an important result, demonstrating that the bed has reached a quasi-equilibrium state under the constant sediment supply conditions used in this validation case. This behavior is consistent with the Parker e-book solution we used for validation, where the continuous sediment supply with the same GSD as the initial bed leads to minimal sorting once equilibrium is reached. The subplot showing temporal evolution of surface and substrate Dsg already helps illustrate this stabilization process.

Regarding the reviewer's suggestions about exploring different sediment supply scenarios and their implications for bed evolution, we share their enthusiasm for this research direction. These investigations would indeed provide valuable insights into the model's capabilities for

studying complex morphodynamic processes. While this model description paper focuses on presenting and validating the component's fundamental capabilities, we are excited to share that we are currently conducting simulations using RiverBedDynamics and will be preparing a companion paper that will thoroughly explore these applications, including the effects of varying sediment supply rates and GSD on bed evolution. We believe these upcoming analyses will demonstrate the full potential of RiverBedDynamics for investigating diverse geomorphic scenarios.

12) *How computational intensive is the application of RiverBedDynamics to real-world problems. Briefly mention about the computation times for different simulations in channel and synthetic watershed, and how feasible is it to apply it to a flood event in a river both spatially and temporally with computational requirements.*

R: We thank the reviewer for this important practical question. The computational intensity of RiverBedDynamics applications largely depends on two key factors: the DEM resolution and the spatial extent of the simulation domain. These parameters control the time step requirements and, consequently, the actual processing time.

For the test cases presented in this paper, simulation times were reasonable: the channel simulations (1500m length with 100m cell size) completed within minutes, while the synthetic watershed simulations (36 km² with 30m resolution) required a few hours on a standard desktop computer. Interestingly, flood event simulations can be computationally more efficient than low-flow conditions because higher water depths allow for larger time steps while maintaining numerical stability, thanks to our coupling with OverlandFlow.

The application of RiverBedDynamics to real-world watersheds is feasible from a computational perspective. However, the main challenge often lies not in computational requirements but in obtaining spatially distributed input data, particularly detailed grain size distributions across the watershed. This limitation is more practical than computational.

We believe that RiverBedDynamics is well-suited for both research applications and practical watershed management scenarios, provided the necessary input data are available.

13) Line 672-677: The procedure of adjusting the water depth after the erosion or deposition is not very clear. How running the model for few internal cycles can give the better water surface elevations over the eroding or depositing cells.

R: We thank the reviewer for requesting clarification about this procedure. We have expanded our explanation of the local water depth correction process in both the main text and Figure 11's caption.

In the main text, we have revised the description to read:

" When bed elevation changes occur at a node, the initial calculation maintains the same water depth but shifts the water surface elevation by the same amount as the bed change. This can create unrealistic spatial jumps in water surface elevation between adjacent nodes that need to be smoothed while preserving discharge. The jumps occur because neighboring nodes still have their original water surface elevations. To address this, OverlandFlow runs for a few internal cycles, to redistribute around the nodes experiencing bed changes and their immediate neighbors (those sharing links). During these cycles, simulation time does not advance, and the correction maintains mass conservation while achieving a smoother water surface profile. Once complete, the corrected water depths are mapped onto the grid, modifying local velocities while preserving discharge. This capability is configured in the driver file, with an example provided in the test case used in section 5.4"

We have also enhanced Figure 11's caption to better illustrate the process. We believe that these revisions provide a clearer explanation of both the physical basis and the numerical implementation of the water depth correction procedure

14) Since many variables and their short representation have been used in the paper and its often confusing what variables are being referred to. It may be better, if the authors can provide the list of variables used and their full names at the end of manuscript.

R:  We thank the reviewer for this helpful suggestion to improve the manuscript's readability. We have added an "Appendix A: Notation" at the end of the manuscript that provides a comprehensive list of all variables used, organized by category (Geometric and Grid Variables, Flow Variables, Sediment Properties, Transport Parameters, etc.). Each variable is presented with its symbol, description, and units.

This organization allows readers to quickly locate and identify variables based on their physical meaning and application within the model. We believe this addition significantly improves the manuscript's clarity and accessibility

**Reviewer #2 – Fergus McNab**

We extend our sincere gratitude to Reviewer #2 for their thoughtful, constructive, and deeply engaged review. Your insightful feedback has significantly strengthened the manuscript, sharpening its focus, enhancing its clarity, and elevating its scientific rigor. The care and expertise evident in your comments—from overarching conceptual critiques to nuanced technical suggestions—have guided critical improvements across the paper.

Your observations about clarifying spatial/temporal scales and potential applications early in the manuscript prompted us to restructure the Introduction, integrate explicit scale definitions, and add concrete examples of research questions the component can address. These revisions not only resolve initial ambiguities but also better position the work within the broader landscape modeling context. Your attention to technical details, from numerical stability considerations to boundary condition explanations, has improved the precision of our methodology and discussion sections. The suggestions to highlight stratigraphy tracking and streamline colloquial phrasing further refined the narrative, ensuring consistency and professionalism.

We particularly appreciate your recognition of the component's innovative aspects, such as the feedback between bed evolution and flood hydrographs, which we now emphasize as a key application in the Introduction.

**General comments**

1) *Anticipated spatial and temporal scales of use*
   *I was unsure, reading the manuscript for the first time, on what spatial and temporal scales and/or resolutions the authors anticipate RiverBedDynamics being applied. Later in the manuscript, the authors do mention that their "mechanistic approach may be better suited for modeling relatively small-time scale processes", that various adjustments/enhancements might be needed "to expand the component's applicability to longer timescales", but also that the component is "flexible enough for short- and long-term simulations". There is some inconsistency here, but my general impression is that the authors have aimed to design a component that can work across time scales, but is probably most suited to short time scales for now. I think this would be good to clarify much earlier in the manuscript, since it influences some of the assumptions made in constructing the model and also the types of problem that can be addressed using it. Not doing so lead to some confusion on my part which I will give some examples of below...*

R: We thank the reviewer for this insightful comment about clarifying the spatial and temporal scales. We agree that being explicit about these aspects early in the manuscript will help readers better understand the component's capabilities and limitations. We have made several revisions to address this:

a) Added a new paragraph in the Introduction:

"RiverBedDynamics is designed with flexibility to operate across various spatial and temporal scales, though it is particularly well-suited for analyzing relatively short-term

processes (hours to years) where detailed grain-scale interactions and non-steady flow conditions are important. While the component can theoretically handle longer timescales, its mechanistic approach to sediment transport and explicit treatment of flow hydraulics makes it computationally intensive for simulations spanning millennia. The spatial resolution can range from fine-scale process representation (meters) to watershed-scale applications (tens of meters), though users must carefully consider the tradeoffs between spatial resolution, computational demands, and the physical processes being represented."

b) Modified the abstract to include:

"This approach provides a comprehensive representation of the complex interactions between flow dynamics and sediment transport in gravel-bedded rivers *at timescales ranging from individual flood events to yearly morphological changes*, enhancing our ability to model landscape evolution across diverse geomorphic settings."

c) Added clarification about the example case spatial resolution:

"We used a 30x30m resolution as it provides a reasonable balance between capturing the key morphological features while maintaining computational efficiency for watershed-scale simulations. While finer resolutions are possible, they become highly computationally demanding for large domains. This resolution allows us to demonstrate the component's capabilities at capturing reach-scale morphodynamics while still enabling watershed-scale analysis."

*2) Potential applications*

*The authors clearly explain some limitations of earlier landscape evolution models, and how their component is a more realistic treatment of process in gravel-bed rivers. But they give few (if any) examples of the kinds of problems they anticipate their component being used to address. The reader is therefore left wondering somewhat why the component is needed (aside from whether or not it is a "major step forward" in a technical sense). It would be great if the authors could give some examples of the kinds of problems that cannot be properly addressed with previous models, but can with this one. I think this would help engage the reader, increase the impact of the paper and inspire future work.*

R: We appreciate this valuable suggestion to better illustrate the practical applications of RiverBedDynamics. We have added several concrete examples of research problems that can be addressed with our component throughout the manuscript, particularly in the Introduction and Discussion sections. These additions help demonstrate the practical utility and potential impact of our work while maintaining technical accuracy.

To address this important comment below we listed all the additions and modifications in our article.

a) Added to Introduction:

" This new modeling capability enables researchers to address several important questions in fluvial geomorphology, such as: (i) How do variations in sediment supply and grain size distributions influence channel morphodynamics during flood events? (ii) What are the feedbacks between bed surface texture evolution and channel adjustment during unsteady flows? (iii) How do spatial patterns of erosion and deposition respond to changing climate

and land use conditions across watershed scales? and (iv) What is the role of grain sorting processes in determining channel stability and sediment connectivity through river networks?"

b) Added to Current capabilities and future enhancements:

" RiverBedDynamics also enables researchers to investigate the impact of changing precipitation patterns on sediment transport and channel morphology, the role of grain size sorting in determining sediment delivery to reservoirs, the effectiveness of different river restoration designs that involve gravel augmentation, and the influence of urbanization on channel stability through changes in both flow and sediment regimes. These applications demonstrate how the component's ability to simulate both erosion and deposition while tracking grain size evolution enables investigation of problems that were difficult to address with previous modeling approaches.

3) *Introduction*
   *I found the Introduction somewhat long-winded and unfocused. Especially at the beginning, there is quite a high level of detail about features of landscape evolution modelling that are not the main focus here. For example, the first paragraph contains quite a lot of detail about different approaches to flow routing, while the second discusses discretization, neither of which are central to the paper. Only in the third paragraph do we start to hear about the specific knowledge gaps that will be addressed by the new component. I suggest restructuring, and potentially shortening, the Introduction, to place more emphasis on the specific issues to be addressed and why there are important. This would also be a good place to clarify already the spatial and temporal scales considered and give some examples of potential applications.*

R:  We thank the reviewer for their constructive suggestions to improve the Introduction's focus and organization. We have made substantial revisions to address these concerns:

    a) We have restructured the Introduction to more quickly highlight the key issues addressed by RiverBedDynamics, while maintaining necessary context about general LEM features. The discussion now flows more logically from existing capabilities and limitations to specific challenges in modeling gravel-bed rivers.
    b) We have improved transitions between sections and eliminated redundant text, making the narrative more concise and focused.
    c) We have added clear statements about the spatial and temporal scales at which RiverBedDynamics is designed to operate, helping readers understand the model's intended applications and limitations.
    d) We have integrated specific examples of research questions that can be addressed with RiverBedDynamics, demonstrating its practical utility for studying fluvial processes.
    e) We have strengthened the connection between the component's capabilities and the existing gaps in landscape evolution modeling.

These revisions maintain all essential technical content while improving readability and better highlighting our contribution to the field. The restructured Introduction provides readers with a clearer understanding of both the need for and capabilities of RiverBedDynamics.

**Line-by-line comments**

1) *L72-74. The authors focus on correctly modelling bed shear stresses as the motivation for modelling the evolution of grain-size distributions. Another motivation might be understanding fluvial deposits (indeed, later on we learn that the component can keep track of the developing stratigraphy).*

R: We thank the reviewer for this suggestion. We agree that understanding fluvial deposits through tracking stratigraphy is an important motivation that we should highlight earlier in the manuscript. We have expanded this section to include both the hydraulic and stratigraphic motivations for modeling grain-size distributions.

We modified the introduction of the paper to early state this motivation. The edited paragraph can be found in the sentences "Very few LEMs include explicit treatment of grain sizes, yet this factor is critical for both accurately simulating sediment transport, channel morphodynamics, and understanding the formation of fluvial deposits. The ability to track grain size …"

2) *L108-110. Here the authors refer to the importance of modelling elevation changes across the entire catchment, not just within channels, and broader landscape interactions. Could some examples of these other processes and interactions be provided? The sediment-transport physics used in the component applies to flowing water, which might not apply outside channels – are other components needed to included other processes?*

R: We appreciate the reviewer's important questions about how our component handles processes across the catchment and beyond channels. We have clarified that while our sediment transport equations can effectively capture overland flow erosion processes (which use similar physics to channel processes), we acknowledge that many hillslope processes require different physical representations. We now more clearly position RiverBedDynamics within Landlab's broader framework, where it can be coupled with other components that handle specific hillslope processes.

In particular, we modified the paragraphs around these lines and added a paragraph that acknowledges the reviewer's comment. The added paragraphs read: "…Our RiverBedDynamics component complements this existing work by addressing different research questions, particularly those requiring continuous grain size distributions and dynamic responses to varying hydrological conditions.

The development of these components within Landlab has progressively enhanced our ability to model different aspects of landscape evolution. However, there remains a need for a component that can simulate the full range of processes occurring in gravel-bed rivers while taking advantage of Landlab's integrative framework. While our component primarily addresses sediment transport by flowing water—applying these physics across both

channelized and unchannelized areas during overland flow events—other geomorphic processes such as soil creep, landsliding, and bank erosion require different physical representations through specialized Landlab components. In particular, the ability to model sediment transport and deposition across varying flow conditions, while tracking both spatial and temporal evolution of grain size distributions, would represent a significant advance in our modeling capabilities."

*3)* and 4) *L113. "What's needed" – colloquial.; L156. "Let's examine" – colloquial. I am not necessarily opposed to this conversational style, but it is hardly used elsewhere in the manuscript, so it stands out here.*

R: We thank the reviewer for noting these inconsistencies in writing style. We agree that maintaining a consistent formal tone throughout the manuscript is preferable and have revised these colloquial phrases.

We modified the sentences to:

"…To address these limitations, a component is required that can:…"

*5)* *This is the only place in the manuscript that snippets of code are given -- in isolation, without more context as to the structure of Landlab components, I don't think it's particularly useful. Consider removing.*

R: We appreciate the reviewer's concern about the code snippets appearing in isolation. However, we believe these snippets serve an important purpose in demonstrating how Landlab's structure enables direct mathematical implementations. Rather than removing them, we propose adding more context to better explain their significance. The new paragraph now reads : "This implementation illustrates a key advantage of Landlab's design: the ability to translate mathematical equations directly into code that operates efficiently across the entire domain. The similarity between Eq. 1 and its implementation demonstrates how Landlab's grid-based structure eliminates the need for explicit iteration over individual cells, making the code both intuitive and computationally efficient…"

*6)* *L182. 'In our implementation...' I was a bit confused by this sentence. The section is about the component, so I initially took 'our implementation' to refer to the component. But then the sentence seems contradictory, since the implementation both utilizes a specific flow-router and can be used with any flow router. Maybe with 'our implementation' you are referring to the wider model you construct and use later in the examples, distinct from the individual component? Consider rephrasing for clarity.*

R: We thank the reviewer for pointing out this ambiguity. The distinction between the component's capabilities and our specific implementation choices needs to be clearer.

Now the paragraph reads: "While RiverBedDynamics is designed to work with any flow solver through Landlab's "plug-and-play" capabilities, in the examples presented in this paper, we demonstrate its use with the OverlandFlow component (Figure 2). This specific choice of flow solver illustrates one possible implementation, but users can couple RiverBedDynamics with other flow routing components available in Landlab depending on their needs"

7) *L186. 'Bed surface grain size properties variables' is a bit of a mouthful -- consider either 'properties' or 'variables'*

R: We thank the reviewer for this editorial suggestion to improve readability.

Now the text reads: "In the first part, RiverBedDynamics processes surface flow and bed surface grain size properties stored in the grid to calculate local shear stress and bed load transport rate."

We simply eliminated the redundant word "variables" while maintaining the full meaning of the sentence.

8) *L200. The term 'unsteady friction slope' is new to me, could a definition be provided?*

R: We thank the reviewer for requesting clarification of this technical term.

We added the following "This formulation is an extension used in shear stress calculations of the commonly used depth-slope product, in which steady and uniform flow conditions are assumed, resulting in $S_{fx} = -\partial\eta/\partial x$ in the x direction. The unsteady friction slope formulation accounts for both spatial variations in flow properties and their temporal changes, including acceleration and deceleration effects that are particularly important during flood events.

9) *L222-236. As I mentioned above, I think the discussion of water depth vs. hydraulic radius for calculating shear stress has implicit assumptions about the scales on which the model will be run. It is good that both options are available, but the emphasis on water depth is justified by a scenario in which the spatial resolution is much finer than that of individual channels (which is not carried out in the later example, where the resolution is 30x30 m)....*

R: We thank the reviewer for this insightful comment about scale assumptions and process representation. We have revised this section to better clarify: (1) how the choice between water depth and hydraulic radius relates to model resolution, (2) the limitations of our component regarding hillslope processes, and (3) how RiverBedDynamics can be integrated with other Landlab components designed for hillslope processes. We now explicitly state that our component focuses on sediment transport by flowing water, while acknowledging that other important hillslope processes like soil creep and mass wasting require different treatment through components such as TransportLengthHillslopeDiffuser and LandslideProbability.

10) *Figure 4. The size differences between the in/out arrows are not immediately obvious visually. An alternative, to make the figure more intuitive, might be to use 2D arrows where both the width and length scale with the sediment flux.*

R: We thank the reviewer for this suggestion about the arrows in Figure 4. We adjusted the arrow sizes to make them more visually distinct, though we note that the primary purpose of the arrows is to illustrate the direction of sediment fluxes across cell faces, as this determines whether erosion or deposition occurs. The left diagram shows how incoming fluxes from multiple directions result in deposition, while the right diagram shows how directional imbalance in fluxes leads to erosion. We also modified the figures caption to include these

changes, now it reads: "Examples of an increasing (left) and decreasing (right) bed surface elevation in time. Arrows indicate the direction of sediment fluxes across cell faces (arrow size adjusted for visibility), with flux directions determining the net bed load transport rate within a cell and consequently dictating erosion or deposition of sediment. In the left diagram…

11) *L312. Could the authors explain their rationale for using an explicit scheme here? In the flow routing, an implicit scheme is used. I assume this choice has implications for the stability of the solution (some later statements related to the example imply that too). It might be useful to give an approximate stability criterion, if one exists, to give readers an idea roughly what combinations of spatial and temporal resolution will be practical when running the model.*

R: We thank the reviewer for requesting clarification of this technical decision. We have expanded that section and include a detailed explanation that reads: " The choice of an explicit scheme for solving the Exner equation (Eq. 22 and 25) was motivated by its simplicity and ease of implementation, particularly when coupled with the OverlandFlow component. While OverlandFlow uses an implicit scheme for flow routing, which is advantageous for maintaining stability in hydrodynamic simulations, …"

By adding this paragraph, we included:

- Rationale for Explicit Scheme: The explicit scheme was chosen for its simplicity and ease of implementation, especially when dealing with sediment transport and bed evolution. It allows for straightforward updates of bed elevation and grain size distribution at each time step, which is critical for capturing the dynamic interactions between flow and sediment transport.
- Stability Considerations: The explicit scheme requires careful control of the time step to avoid numerical instability. The CFL condition for sediment transport provides a useful guideline, but in practice, the time step must also account for the coupled nature of the system, where rapid changes in bed elevation can affect flow conditions and vice versa.
- Practical Implications: Users should be aware that the explicit scheme may require smaller time steps compared to implicit methods, particularly in scenarios with high sediment transport rates or steep gradients. This is why we recommend starting with conservative time steps and adjusting based on simulation results.

12) *L320-321. I was surprised that boundary conditions are only needed at the outlet – I am used to solving Exner-type equations in one spatial dimension (i.e. longitudinal profiles), where we definitely do need boundary conditions at both ends of the domain. Is there an implicit assumption here, e.g. no sediment flux (zero gradient) over the edges of the watershed? If so, that might be perfectly reasonable, but it would be good to state it explicitly.*

R: We thank the reviewer for this important point about boundary conditions. We should have been more explicit that the boundary conditions mentioned in the text specifically refer to bed surface elevation. Sediment fluxes can be flexibly specified at any node in the domain, including inlet and outlet links as well as internal nodes. We have clarified this distinction in the revised text.

We added "However, it's important to note that sediment fluxes can be independently specified at any location in the domain, including inlet boundaries, outlet boundaries, and

internal nodes. This flexible approach allows users to implement various sediment supply scenarios and internal sediment sources or sinks."

13) *The default boundary condition for the outlet is "zero gradient". Does this imply zero sediment flux out of the watershed? That seems strange to me if so, and the fixed elevation option (base level) would be more familiar. Could a brief description of the physical meaning of the zero gradient option be included? Also, I don't follow the subsequent description of the implementation of the zero gradient option. Please reconsider the phrasing there.*

R: Thank you for raising this important point about the zero gradient boundary condition. We agree that our explanation needs clarification. The zero gradient condition for bed elevation does not imply zero sediment flux - rather, it allows sediment to freely pass through the outlet based on local transport capacity. We have revised the text to better explain the physical meaning and implementation of this boundary condition.

We added/modified the text which now reads: "The zero-gradient condition allows the bed elevation at the outlet to evolve naturally based on upstream conditions, maintaining the same slope as the immediately upstream reach. This approach permits sediment to be transported through the outlet based on local transport capacity, rather than imposing an artificial constraint on sediment flux. The fixed-value condition, alternatively, maintains a constant bed elevation at the outlet, which can be useful when modeling systems with known base levels."

"In practice, the zero-gradient condition is implemented using user-provided methods for the watershed outlet boundary condition (e.g., set_watershed_boundary_condition_outlet_id, set_watershed_boundary_condition) to identify all connecting nodes and links upstream of the outlet. At the end of the bed elevation update routine, RiverBedDynamics matches the bed elevation of outlet nodes to their immediate upstream neighbors, ensuring smooth transitions and preventing artificial disruptions to sediment transport at the boundary. When other types of boundary…"

14) *L331. I was initially unsure why the different sediment layers needed to be defined and tracked. It becomes clear later (e.g. L352-352), but only after going through quite a lot of details. Perhaps the start of this paragraph can be rephrased to make the motivation/necessity clear from the beginning.*

R: We thank the reviewer for this helpful suggestion. We agree that providing the motivation for tracking different sediment layers upfront would improve readability. We have revised the text to explain the importance of these layers before diving into technical details.

We modified the text and now that section reads: "Sediment mass entering or leaving a cell can not only alter the bed surface elevation but also the bed GSD. We represent the evolution of the surface and substrate GSD by means of three layers (bed load, surface, and substrate) to capture key physical processes in gravel-bed rivers. This multi-layer approach is essential for accurately modeling: (1) the interaction between flow and the active surface layer where sediment exchange occurs, (2) the development of bed armoring and sorting patterns, and (3) the preservation of depositional history in the substrate, which influences future bed evolution. The bed load layer represents…"

15) *The stratigraphy tracking is interesting and potentially useful in geological applications – it could be mentioned earlier (e.g. in the Introduction) as part of the general motivation behind the component.*

R: Thank you for this suggestion. We agree that highlighting the stratigraphy tracking capability earlier would better showcase one of the component's unique features. We have added text in the Introduction to emphasize this capability and its relevance for geological and geomorphological applications.

At the introduction we added and edited the text, which now reads: " To address this gap, we developed RiverBedDynamics, a new Landlab component designed to simulate the evolution of gravel-bed streams in a continuum model, allowing for the integration of the channel with other geomorphological processes using the Landlab platform. By coupling RiverBedDynamics with OverlandFlow, we enable the simulation of non-steady flow conditions, which is crucial for understanding the complex dynamics of gravel-bed rivers. The component addresses the limitations of existing models by incorporating grain size evolution, non-steady flow conditions, and watershed-wide predictions of bed surface changes. A key feature of our component is its ability to track and preserve stratigraphic information, enabling investigations of both short-term morphodynamics and longer-term geological processes. This stratigraphy tracking capability allows researchers to study how temporal variations in flow and sediment transport conditions are recorded in depositional sequences, providing insights into the geological history of river systems.

In this article, we introduce RiverBedDynamics and demonstrate its capabilities in modeling gravel-bed river evolution within a landscape context. Through a series of tests and applications, we illustrate how RiverBedDynamics can capture key aspects (e.g., sediment transport dynamics, grain size evolution, erosion and deposition processes, and watershed-scale bed surface changes) of landscape evolution, particularly…"

16) *L410. The previous section was also number four.*

R: Thank you for catching this numbering error. You are correct - there was a duplicate section number. We have corrected the section numbering throughout the manuscript to ensure proper sequential numbering of all sections.

17) *Can you comment more on the stability issue? Is there a predictable stability criterion (e.g. related to the explicit solution to the elevation equation)? Or does the instability come more from the coupling of the two equations, and need to be found by trial and error? In either case, this would be useful practical information for potential users.*

R: Thank you for this important question about stability. We have expanded the text to better explain both the theoretical stability criteria and practical considerations arising from the coupled nature of the system.

We added the following text: "This stability constraint arises from two sources. First, the explicit solution of the Exner equation imposes a theoretical stability criterion of $\Delta t \leq (1-\lambda\_p)\Delta x \Delta y / q\_b$ . Second, the coupling between flow and bed evolution introduces additional stability requirements, particularly during rapid morphological changes when there is strong feedback between bed elevation changes and flow conditions. In practice, we found that time

steps satisfying the theoretical criterion may still need to be reduced by a factor of 2-5 to maintain stability in the coupled system, especially in areas of high sediment transport or rapid bed evolution. For the conditions in our test case (spatial resolution of 25-100 m, typical bed load transport rates), this resulted in stable time steps of 1-5 seconds. Users implementing different conditions should start with conservative time steps and adjust based on stability monitoring."

18) *L504-506. I am curious about the physical reason behind the (quite significant) differences in equilibrium slope for the various models. Could you briefly comment on that? In general, we don't learn much about the differences between the parameterizations (other than that they deal with grain size in different ways). I appreciate it is not the purpose of the paper to review these models, but some basic information might be useful (also above where the models are introduced), to aid readers in choosing an appropriate parameterization for their purposes.*

R:  Thank you for this insightful comment. We agree that explaining the physical reasons for differences in equilibrium slopes would help readers better understand the models' behaviors and support their choice of parameterization. We have added text explaining these differences and expanded the earlier model descriptions.

19) *L514-518. Similarly, it is interesting to see the downstream part fining and then coarsening – could you briefly comment on the physical reason for that?*

R:  We appreciate this suggestion to clarify the physical mechanisms. We have added text explaining how the observed pattern emerges from the interaction between selective transport processes and local transport capacity. The sequential fining and coarsening reflects both the initial mobility differences between grain sizes and the subsequent development of armor layers through winnowing processes.

We added: "The upstream portion of the channel initially coarsened compared to the original bed grain size because the supply of upstream finer sediment was less than the transport capacity of this sized material. Such finer sediment was therefore eroded from the most upstream cells, supplying finer sediment downstream. This upstream erosion and preferential transport of the finer sediment led to the pattern of downstream fining in the beginning of the simulation (e.g., 1-10 days). However, this pattern was temporary. Given that the upstream supply of finer sediment at the simulation boundary did not replace the removed fine bed material at a sufficient rate, fine sediment was progressively winnowed throughout the entire model domain through downstream transport and vertical sorting into the subsurface. As this winnowing process continued, the initial downstream fining pattern was gradually replaced by widespread surface armoring, where coarser grains became concentrated at the surface, resulting in systematic coarsening across the domain at later timesteps. This sequence illustrates the dynamic interaction between selective transport, sediment supply, and local hydraulic conditions during bed adjustment, showing how temporal changes in sediment availability can shift the dominant grain sorting pattern.

20) *L555-556. I would expect the 30x30 m resolution to be much larger than most (all?) channels in a 6x6 km watershed. Is water depth or hydraulic radius used for the shear stress calculation? This might be a useful example to help readers think through which of the two*

*options is appropriate given the resolution/scale of a given problem. Are there practical limitations that meant you chose this relatively coarse resolution?*

R: Thank you for raising this important methodological point. We have expanded our discussion of resolution choice to address both the practical limitations and its implications for shear stress calculations. We explain why water depth is more appropriate than hydraulic radius at this resolution, while acknowledging the trade-off between capturing detailed channel geometry and enabling watershed-scale simulations. The added text also helps readers understand how to choose between water depth and hydraulic radius calculations based on their specific application and grid resolution. Reviewer #1 had a similar comment and we addressed both suggestions in one new paragraph.

The new paragraph reads "For this watershed-scale application, we selected a 30 x 30 m grid spacing as it balances computational efficiency with the ability to capture key morphological features. While finer resolutions are possible, they become highly computationally demanding for large domains. This choice allows us to demonstrate the component's capabilities at capturing reach-scale morphodynamics while still enabling watershed-scale analysis. In a watershed of this size, grid cells are typically wider than natural channel widths, which influenced our approach to shear stress calculations. Specifically, we use water depth rather than hydraulic radius for two main reasons. First, when a cell is wider than the natural channel, most cell boundaries interface with other water-filled cells rather than channel banks, making hydraulic radius less representative of actual flow conditions. Second, our grid cells capture both channelized and overland flow areas, where the concept of hydraulic radius is less applicable. While this approach sacrifices some detail in channel geometry, it provides reasonable estimates of reach-scale sediment transport across the watershed."

21) *L558-559. "both having uniform rainfall and the same total volume of water precipitated (24 mm) over all cells" – quite a complex phrase, do you just mean spatially uniform rainfall?*

R: Thank you for suggesting this clarification. We have simplified the description of the rainfall conditions to more clearly state that the rainfall was spatially uniform across the domain, while maintaining the important information about total precipitation volume.

22) *L578-579. It is cool to see the feedback between the bed-elevation changes and the hydrograph – this would be an example of the kind of problem that can be explored with the component, that you could mention already in the Introduction.*

R: Thank you for this excellent suggestion. We have added the feedback between bed elevation changes and flood hydrographs as one of the key examples of problems that can be explored with RiverBedDynamics in the Introduction. This helps readers better understand the component's capabilities from the outset and provides a clearer link to the results we present later in the paper.

We added the following text to the introduction: "This advancement enables researchers to address several important questions in fluvial geomorphology, such as: (i) How do variations in sediment supply and grain size distributions influence channel morphodynamics during flood events? (ii) What are the feedbacks between bed surface evolution and flood wave propagation through river networks? (iii) How do spatial patterns of erosion and deposition respond to changing climate and land use conditions across watershed scales? and (iv) What

is the role of grain sorting processes in determining channel stability and sediment connectivity through river networks?"

23) *Figure 10. The inset hydrographs seem to have flipped vertical axes – is that deliberate?*

R: Thank you for noting this. Yes, the inverted vertical axis for rainfall intensity in the inset hydrographs is deliberate. We chose this presentation format to better illustrate the cascading relationship between precipitation input, discharge, and bed elevation response. We believe that this layout allows readers to more easily trace how the rainfall signal propagates through the system and influences both flow and morphological changes. We have added a note in the figure caption to clarify this intentional choice.

24) *L636-637. "Second, all our test cases involved channels without macro-roughness elements such as large boulders, vegetation, or any type of flow obstructions that can significantly alter the flow direction." This kind of comment suggests to me that the authors expect the component to be run at very fine resolutions, which contrasts with the relatively coarse resolution of the example – this is the kind of thing that could be cleared up by explicitly stating somewhere the scales you are thinking about, and also explaining your choice of resolution in the example.*

R: Thank you for highlighting this important point about scale compatibility. We have expanded the text to clarify that while the component can theoretically operate at finer resolutions capable of resolving macro-roughness elements, practical watershed-scale applications often require coarser grids. We explain that this limitation creates a need for future development of sub-grid scale parameterizations to account for these features in coarser resolution simulations. This addition helps readers better understand the component's current capabilities and limitations across different spatial scales.

We added the following: "While our component can theoretically be applied across a range of spatial scales (from meters to tens of meters), the computational demands of watershed-scale applications often necessitate coarser resolutions. In such cases, the effects of sub-grid scale features like macro-roughness elements would need to be parameterized rather than explicitly represented. Future development could incorporate methods to account for these small-scale effects within coarser resolution simulations"

25) *L664-665. "The coupled OverlandFlow-RiverBedDynamics approach in our LEM employs a decoupled method to solve for river bed evolution." The authors refer to the flow routing and surface evolution as being "coupled" or in "continuous feedback" (L181) at various points in the manuscript, but here say that they are also "decoupled". I understand what is meant (that over many time steps the two are coupled/in feedback, but in a single time step they are applied sequentially), but this wording might cause confusion. Consider revising (here and elsewhere in the manuscript).*

R: Thank you for identifying this potential source of confusion. We have revised the text to more precisely describe our computational approach, distinguishing between the physical coupling of processes and the sequential solution strategy used within each time step. We've replaced potentially ambiguous uses of 'decoupled' with clearer descriptions of the numerical

implementation while maintaining accurate representation of the physical feedbacks in the system.

The specific sentence the reviewer mentioned now reads: "The OverlandFlow-RiverBedDynamics approach in our LEM employs a sequential solution strategy (Cao et al., 2002; Colombini & Stocchino, 2005). This means that the governing equations are solved separately and serially. Essentially, the flow is "paused" while the RiverBedDynamics component solves the Exner equation during a given time step (Figure 11 a and b). While the components interact through continuous feedback over multiple time steps, their equations are solved one after another within each individual step…"

Other instances that were corrected now read:

Abstract: "LEMs typically comprise at least two interacting components: a flow hydraulics solver..."

Model description: "Our component was designed to work in conjunction with a flow solver such that continuous feedback between surface flow and river bed properties determines the behavior of the system. While RiverBedDynamics is compatible with any flow solver through Landlab's "plug-and-play" capabilities, in the examples presented in this paper, we demonstrate its use with the OverlandFlow component (Figure 2). This specific choice of flow solver illustrates one possible implementation, but users can integrate RiverBedDynamics with other flow routing components available in Landlab depending on their needs…"

Conclusion: "Integrating RiverBedDynamics with OverlandFlow has created a LEM capable of..."

26) *L671-676. I don't understand the correction being applied here. The authors write that "only the water surface elevation is affected after a change in bed surface elevation, not water depth or discharge", but in the illustration, it is clear that different water depths are obtained at the end of the correction. Is the change in bed elevation re-calculated with the new water depths? If so, it is not mentioned. If not, I'm not sure what the purpose of the correction is in the first place.*

R: Thank you for this important observation. We have revised the text to clarify the water surface correction process. The apparent contradiction arose from imprecise language in our original description. We now explain that while discharge is preserved, water depths do change during the correction phase as OverlandFlow redistributes water around modified bed elevations to maintain a physically realistic water surface profile. The bed elevation changes themselves are not recalculated - rather, the correction ensures mass conservation and numerical stability while transitioning to the new bed configuration. We have added implementation details and references to example code.

Reviewer#1 made a similar comment regarding this figure. We modifies the text in that section and now it reads:
"When bed elevation changes occur, only the water surface elevation is initially affected, while discharge remains constant. This may create a local discontinuity in water surface elevation that needs to be adjusted. To address this, OverlandFlow runs for a few internal cycles, to redistribute around the nodes experiencing bed changes and their immediate neighbors (those sharing links). During these cycles, simulation time does not advance, and the correction maintains mass conservation while achieving a smoother water surface profile.

Once complete, the corrected water depths are mapped onto the grid, modifying local velocities while preserving discharge. This capability is configured in the driver file, with an example provided in the test case used in section 5.4."

The caption in Figure 11 was also slightly modified to enhance this explanation.

27) *L710-711. "Additionally, incorporating bank erosion and channel migration capabilities could improve predictions and make long-term simulations more realistic." This seems quite a big leap to me, since the current framework uses a uniform gird and does not explicitly distinguish channels and their widths from the rest of the landscape. Perhaps the authors could expand on how this might be achieved, or give some references to studies that attempt a similar thing. Or consider removing.*

R: Thank you for raising this important point. You are correct that proposing bank erosion and channel migration capabilities without addressing their implementation challenges in our uniform grid framework required further clarification. In response, we have revised the text to explicitly acknowledge the limitations of our current grid-scale approach. We now reference established methodologies like Van De Wiel et al. (2007) to illustrate how sub-grid parameterizations could bridge these gaps, enabling representation of channel migration processes within our framework's constraints.

Additionally, we have expanded the discussion to include exploring integration with Landlab's unstructured grid handling system (ModelGrid). Adopting Landlab's tools could mitigate grid-scale limitations by enabling more flexible representation of irregular river geometries and heterogeneous topography, while retaining computational efficiency. This dual-path approach—leveraging sub-grid parameterizations in the short term and grid structural advancements in the longer term—provides readers with a clearer roadmap for future development, balancing technical feasibility with scientific research capabilities.

The revised text in the manuscript now reads:

" Additionally, future development could incorporate sub-grid parameterizations of bank erosion and channel migration processes. While our uniform grid framework cannot explicitly resolve channel widths, approaches similar to those developed by Van De Wiel et al. (2007) could inform how such processes might be represented through carefully designed parameterizations that account for grid-scale limitations. RiverBedDynamics could also benefit from adopting Landlab's unstructured grid handling system (ModelGrid). By extending the component's formulation to leverage Landlab's gridding library, the model could better represent irregular river geometries and heterogeneous topographic features in the surrounding landscape. To expand…

---

## Author Response (AR2)

**Responses to Reviewer's Comments – Final iteration**
**RiverBedDynamics v1.0: A Landlab component for computing two-dimensional sediment transport and river bed evolution**
By: Angel D. Monsalve, Samuel R. Anderson, Nicole M. Gasparini, Elowyn M. Yager

**Reviewer #2**

We sincerely thank Reviewer #2 for his continued engagement with our manuscript and for providing these additional comments that have helped refine our paper and enhance its quality. Your thoughtful feedback throughout the review process has been instrumental in improving the clarity, accessibility, and scientific rigor of our work. We are grateful for your attention to important details like the justification for our shear stress calculations and boundary conditions. Below, we address your specific comments in this final round of review.

**Comments**

*Comment #1:*
First, the authors have added some information justifying their choice of defining the bed shear stress using water depth rather than hydraulic radius as a default and in their specific examples. The additional information is appreciated, however, I think there is still some potential for confusion. Where the issue is first introduced (L374-384), the authors argue that when cell sizes are smaller than channel widths, most cells will not contain river banks, so resistance by the banks can be safely ignored (hence using water depth do define the shear stress). However, later, in the context of the example (L891-893), the authors instead justify using the water depth because the cell sizes are much larger than a channel width; then, most cells will include a channel, so most cells are adjacent to other channel cells, and again the banks should be neglected. So it seems that the authors recommend using water depth both when cells are much smaller than channels, and when they are much larger, and the hydraulic radius should only be used in the specific case where the cells are approximately the width of a channel? If so, I think it would make sense to collect the reasoning together and state it explicitly, somewhere around Eq. 5. Then this discussion can be briefly referred back to at the specific example. I think that would help best help readers make an informed choice for their own implementations.

*Response:*
We appreciate the reviewer's careful reading and helpful suggestion to clarify our reasoning for using water depth as the default for shear stress calculations. To address this comment, we have revised the explanation near Eq. 5 and the discussion in the watershed-scale example to ensure consistency and avoid potential confusion.

In the revised text, we explicitly state that the choice of using water depth rather than hydraulic radius depends on the spatial resolution relative to channel width. Specifically, we use water depth as the default for the following reasons:

1. When cell sizes are much smaller than channel widths, most cells represent the channel interior and do not contain river banks. In this case, the hydraulic radius effectively

becomes equivalent to water depth, making the additional complexity of hydraulic radius unnecessary.

2. When cell sizes are much larger than channel widths, a single cell contains both the channel and surrounding floodplain. In this case, most cell boundaries interface with other water-filled cells rather than channel banks, meaning that the hydraulic radius is less representative of actual flow conditions.

3. Our model simulates both channelized flow and overland flow using the same governing equations. Since hydraulic radius relies on the presence of defined banks, it is not applicable in overland flow conditions, further justifying the use of water depth as the default option.

We have consolidated this reasoning in the section around Eq. 5, as suggested, and provided a brief reference back to this explanation in the watershed-scale example. Additionally, we clarify that the hydraulic radius approach is most appropriate when the cell size closely matches the channel width, as this is the scenario where bank roughness significantly influences shear stress calculations.

These revisions ensure that our justification is clearly stated and consistently applied throughout the manuscript. Thank you again for your insightful comment, which helped improve the clarity and accessibility of our methodology.

*Comment # 2:*
Second, the authors now include extra explanation of their 'zero-gradient' boundary condition option for the domain outlet (L515-517). I now understand that the slope is set to match the slope at points immediately upstream, and so clearly does not result in no sediment flux out of the domain. However, in this context, the name 'zero-gradient' does not make much sense to me. I suppose the point is there no gradient in slope? But since slope is itself a gradient, I think this could cause confusion (as it did for me). If this is a standard term. I suppose it has to be retained. But if it is something the authors defined themselves, I suggest revising.

*Response:*
We appreciate the reviewer's careful consideration of our description of the zero-gradient boundary condition. We understand the concern that the name zero-gradient might be misleading since slope itself is a gradient, which could cause confusion.

To clarify, we have revised the text to explicitly state that the zero-gradient condition maintains the bed surface elevation slope of the immediately upstream reach. This ensures that sediment transport continues naturally without imposing artificial constraints at the outlet. We also explicitly note that the net sediment exchange at boundary cells is identical to that upstream of the outlet, reinforcing that sediment flux is maintained through the domain boundary.

We have retained the term zero-gradient because it aligns with its usage in our model implementation and is consistent with common numerical modeling practices. However, we have clarified its meaning within the text to ensure that readers can readily interpret it in the intended context.

We believe these revisions address the reviewer's concern while improving the clarity and accuracy of the manuscript. We appreciate the insightful feedback and the opportunity to refine our explanation.

---

## Author Response (AR3)

**Changes on Final iteration**

**RiverBedDynamics v1.0: A Landlab component for computing two-dimensional sediment transport and river bed evolution**

By: Angel D. Monsalve, Samuel R. Anderson, Nicole M. Gasparini, Elowyn M. Yager

**Dear editor**

We are extremely happy about the recent news we received about our paper egusphere-2024-3390 Title: RiverBedDynamics v1.0: A Landlab component for computing two-dimensional sediment transport and river bed evolution, being accepted for publication in the EGU journal **Geoscientific Model Development.**

In this version we did not edit nor changed any content in the manuscript, except we updated the reference manager style (we use Zotero) to that of Copernicus and made sure that all references are properly listed in the reference section. We also made sure that the style is consistent throughout the paper and there were no articles with capital letter in each word.